# Spontaneous base flipping helps drive Nsp15's preferences in double stranded RNA substrates

Zoe M. Wright[1] ✉, Kevin John Butay[2,5], Juno M. Krahn[2], Isha M. Wilson[1,6], Scott A. Gabel[2], Eugene F. DeRose [2], Israa S. Hissein[1], Jason G. Williams[3], Mario J. Borgnia [2], Meredith N. Frazier[1,4], Geoffrey A. Mueller [2] & Robin E. Stanley [1] ✉

Coronaviruses evade detection by the host immune system with the help of the endoribonuclease Nsp15, which regulates levels of viral double stranded RNA by cleaving 3′ of uridine (U). While prior structural data shows that to cleave double stranded RNA, Nsp15's target U must be flipped out of the helix, it is not yet understood whether Nsp15 initiates flipping or captures spontaneously flipped bases. We address this gap by designing fluorinated double stranded RNA substrates that allow us to directly relate a U's sequence context to both its tendency to spontaneously flip and its susceptibility to cleavage by Nsp15. Through a combination of nuclease assays, $^{19}$F NMR spectroscopy, mass spectrometry, and single particle cryo-EM, we determine that Nsp15 acts most efficiently on unpaired Us, particularly those that are already flipped. Across sequence contexts, we find Nsp15's cleavage efficiency to be directly related to that U's tendency to spontaneously flip. Overall, our findings unify previous characterizations of Nsp15's cleavage preferences, and suggest that activity of Nsp15 during infection is partially driven by bulged or otherwise relatively accessible Us that appear at strategic positions in the viral RNA.

Coronaviruses including SARS-CoV-2 encode for a number of non-structural proteins (Nsps) that aid the virus in replication, transcription, and evasion of the host immune system[1]. One such protein, Nsp15, has been shown to have a direct and significant influence on the severity and duration of coronaviral infection in animal models[2,3]. During infection, Nsp15 functions to limit evidence of viral replication by cleaving RNA 3′ of uridine (U). This prevents the accumulation of long double-stranded RNA (dsRNA) that would otherwise be recognized by multiple host dsRNA-sensing proteins involved in immune

response to viral infection, including MDA5[2,4]. In coronaviruses, Nsp15 is a homohexamer with features that are well-conserved across viral species and variants[5–7]. Each of Nsp15's six protomers contain a C-terminal uridine-specific endonuclease (endoU) domain, with the rest of the protomer controlling oligomerization and supporting RNA binding[7–9]. The endoU domain is a unique genetic hallmark shared across not only coronaviruses but also most nidoviruses[5], with only a few select orthologs in mammalian cells isolated primarily in placental and tumor tissues[10]. This makes Nsp15 an interesting target for

[1]Molecular and Cellular Biology Laboratory, National Institute of Environmental Health Sciences, National Institutes of Health, Department of Health and Human Services, 111 T. W. Alexander Drive, Research Triangle Park, NC 27709, USA. [2]Genome Integrity and Structural Biology Laboratory, National Institute of Environmental Health Sciences, National Institutes of Health, Department of Health and Human Services, 111 T. W. Alexander Drive, Research Triangle Park, NC 27709, USA. [3]Epigenetics and RNA Biology Laboratory, National Institute of Environmental Health Sciences, National Institutes of Health, Department of Health and Human Services, 111 T. W. Alexander Drive, Research Triangle Park, NC 27709, USA. [4]Department of Chemistry and Biochemistry, College of Charleston, 66 George St, Charleston, SC 29424, USA. [5]Present address: Department of Biochemistry, Duke University, Durham, NC 27710, USA. [6]Present address: Howard University College of Medicine, Washington, DC 20059, USA. ✉e-mail: zoe.wright@nih.gov; robin.stanley@nih.gov

therapeutic development, with potential additional impacts for zoonotic and agriculturally-relevant animal disease[5,6].

Coronavirus Nsp15 has been shown to act with broad specificity, cleaving 3′ of U bases in both single-stranded RNA (ssRNA) and dsRNA – though cleavage patterns for dsRNA substrates differ from those of ssRNA with the same sequences[8], for reasons that have not been fully established. In short ssRNA oligos, Nsp15's cleavage activity on a given U is driven by the base 3′ of the U, with U ↓ A and U ↓ G being preferred over U ↓ C for SARS-CoV-2[11]. In dsRNA, by contrast, Nsp15 has been reported to favor a variety of different dsRNA substrates: A/U rich regions[12], stem loop-forming polyU regions in the viral negative strand genomic RNA[13], and UUU regions near the ends of short dsRNA oligos[8]; U's that cannot or do not participate in Watson-Crick H-bonding[14,15]; and/or Us in structured RNA with lower thermodynamic stability[12,15]. In murine cells infected with a mouse coronavirus (MHV), Nsp15 cleavage sites were identified at U ↓ A and C ↓ A sequences primarily within the viral positive strand genomic RNA[16] – though the influence of RNA secondary structure is not clear for all cleavage sites. Nsp15's cleavage efficiency is affected by the presence of $Mn^{2+}$ ions, with cleavage of ssRNA oligonucleotides being accelerated at high concentrations of $Mn^{2+}$[12,17]. While the susceptibility of a particular U to cleavage by Nsp15 does appear to be influenced by its sequence and complement context, Nsp15 does not rely on sequence-specific interactions to bind dsRNA. Like many dsRNA-binding proteins[18], Nsp15 primarily associates with the sugar-phosphate backbone of dsRNA[8], and its binding affinity is not significantly changed by the presence or absence of U in the dsRNA sequence[15].

Additionally, prior structural data of Nsp15-bound dsRNA shows that Nsp15 utilizes a base flipping mechanism to cleave dsRNA[8]. It is not known to what extent Nsp15 passively relies on spontaneous base flipping to identify its targets versus actively promoting base flipping. Previous modeling and structural investigations in both DNA and dsRNA provide important baselines for understanding the energetics of base flipping[19–25], although it remains challenging to predict a particular base's susceptibility to spontaneous flipping in a given sequence of RNA[23]. The process of base flipping can be broken into two energetic requirements: base unstacking and base extrusion[21,26,27]. The energy of base stacking is a major contributor to the stability of a helix[28] and has been thoroughly characterized for simple dinucleotide systems in both DNA and RNA, with pyrimidines (U/ T and C) contributing less stacking energy than purines (A and G)[19,20]. Base extrusion, or rotation of the base out and away from the helix, is less energetically demanding than base unstacking at relatively low angles of rotation, being stabilized by the formation of new, transient H-bonds with nearby bases, sugars, and/or phosphates[19,21,22]; beyond 30 – 50° of rotation, however, bases face a steeper energy barrier to continued rotation. Spontaneous base flipping events that are confined to modest degrees of rotation are therefore much more common than events leading to more extreme rotation[23].

Understanding base flipping in dsRNA beyond simple few-nucleotide systems presents a challenge due to the diversity and dynamicity of the secondary and tertiary structures accessible to RNA[28]. Even in ssRNA with no reported secondary structure, base stacking can strongly influence the accessibility of individual bases[28]. This gap is starting to be bridged by structural studies of RNA that capture information about base dynamics via a variety of techniques: chemical probing such as selective 2′-hydroxyl acylation analyzed by primer extension (SHAPE)[29,30], Nuclear Magnetic Resonance (NMR)[31], computational modeling[30,32], serial femtosecond crystallography[33–35], and increasingly, cryo-electron microscopy (cryo-EM)[36,37]. Still, literature relating spontaneous base flipping in RNA to enzymatic efficiency – for any ribonuclease – remains sparse[22,23,38–40].

Here, we address this gap by characterizing the effect a U's sequence context in dsRNA has on both (1) Nsp15 cleavage activity (via nuclease assays) and (2) spontaneous extrusion of the U from the dsRNA helix in the absence of Nsp15 (via 1D $^{19}$F NMR spectroscopy). To accomplish these experiments, we designed a set of synthetic fluorinated dsRNA substrates that allow us to directly compare the susceptibility of a particular U to either cleavage or spontaneous extrusion in different neighbor and complement contexts. We also determined a cryo-EM structure of Nsp15 bound to dsRNA containing one highly preferred U, which offers structural information about conformational features in dsRNA that are associated with accelerated cleavage. Characterizing the context-dependence of dsRNA cleavage by Nsp15 sheds light on the factors that influence which sites in viral RNA are targeted by Nsp15, deepening our understanding of the role played by Nsp15 during coronaviral infection. More broadly, this work points to a relationship between enzymatic efficiency and spontaneous base flipping for proteins that act on duplexed nucleic acids via a base flipping mechanism.

## Results

### Nsp15 preferentially cleaves unpaired Us in dsRNA

In order for a U in dsRNA to engage with Nsp15's active site, the U must be flipped out of the helix. Base flipping in dsRNA is both more complex and less explored relative to DNA[23,31,32], making it difficult to directly predict how sequence context and secondary structure will affect a given base's tendency to spontaneously flip. Further, Nsp15's contributions to base flipping have not been established. Thus, we aimed to explore the relationship between sequence context, base flipping, and susceptibility to cleavage by Nsp15.

First, we directly probed the effect of a U's complement on its cleavage by Nsp15 by designing a set of five 35 nt dsRNA substrates where the U at position 19 was either matched (U-A), mismatched (U•U′, U•C), engaged in a wobble pair (U•G), or unpaired in a 1 nt bulge (Fig. 1A, Supplementary Table 1). The sequence surrounding U19 was adapted from the nucleocapsid protein transcriptional regulatory sequence (TRS-N) in the SARS-CoV-2 genome, a region known to form dsRNA[1,41,42]. Moreover, Nsp15 cleavage products have been detected within TRS elements in Mouse Hepatitis Virus (MHV) infected cells, suggesting that TRS sequences are among the physiological targets of Nsp15[16]. To facilitate detection of cleavage, the target strand (containing U19) was labeled at the 5′ and 3′ ends by Cy5 and Fluorescein dyes respectively, while the complement strand was unlabeled. To directly compare cleavage between substrates, all Us other than U19 in the target strand were replaced by 2′-fluoro-uridines (2′-F-Us), which can be sampled by Nsp15's active site but cannot undergo the chemical reaction necessary for cleavage. For the U•U′ mismatch, we also substituted the complementary U for an uncleavable 2′-F-U (U′). The 2′-F modification slightly increases the melting temperature of the dsRNA but does not alter its secondary structure[43]. This substrate design effectively limits detectable cleavage events to a single position in the target strand, the central U19 (Fig. 1A). We incubated each of the five synthetic dsRNA substrates at ten times molar excess to purified hexameric Nsp15 and collected aliquots of the cleavage reaction at multiple points over the course of an hour, based on protocols previously optimized to show differences between substrates over this timeframe[8,11]. We used denaturing PAGE to visualize the extent of cleavage at each time point (Fig. 1B), and quantified the intensity of the uncleaved RNA band over time (Fig. 1C).

From this set of reactions, we found that the unpaired U was especially susceptible to cleavage by Nsp15, achieving 59 ± 3% cleavage after 1 h. By contrast, the U•G wobble pair was exceptionally resistant to cleavage, achieving only 7 ± 5% cleavage after 1 h. The U-A match, U•U′ mismatch, and U•C mismatch were intermediate, achieving 28 ± 7%, 25 ± 5%, and 21 ± 7% cleavage respectively. A one-way ANOVA followed by a Tukey HSD post-hoc test indicates that cleavage of the unpaired U is significantly greater than all other substrates ($p = 0.0002, 0.0002, 0.0001, 0.0000$ for unpaired U versus U-A, U•U′, U•C, U•G respectively with a 95% confidence interval) and cleavage of

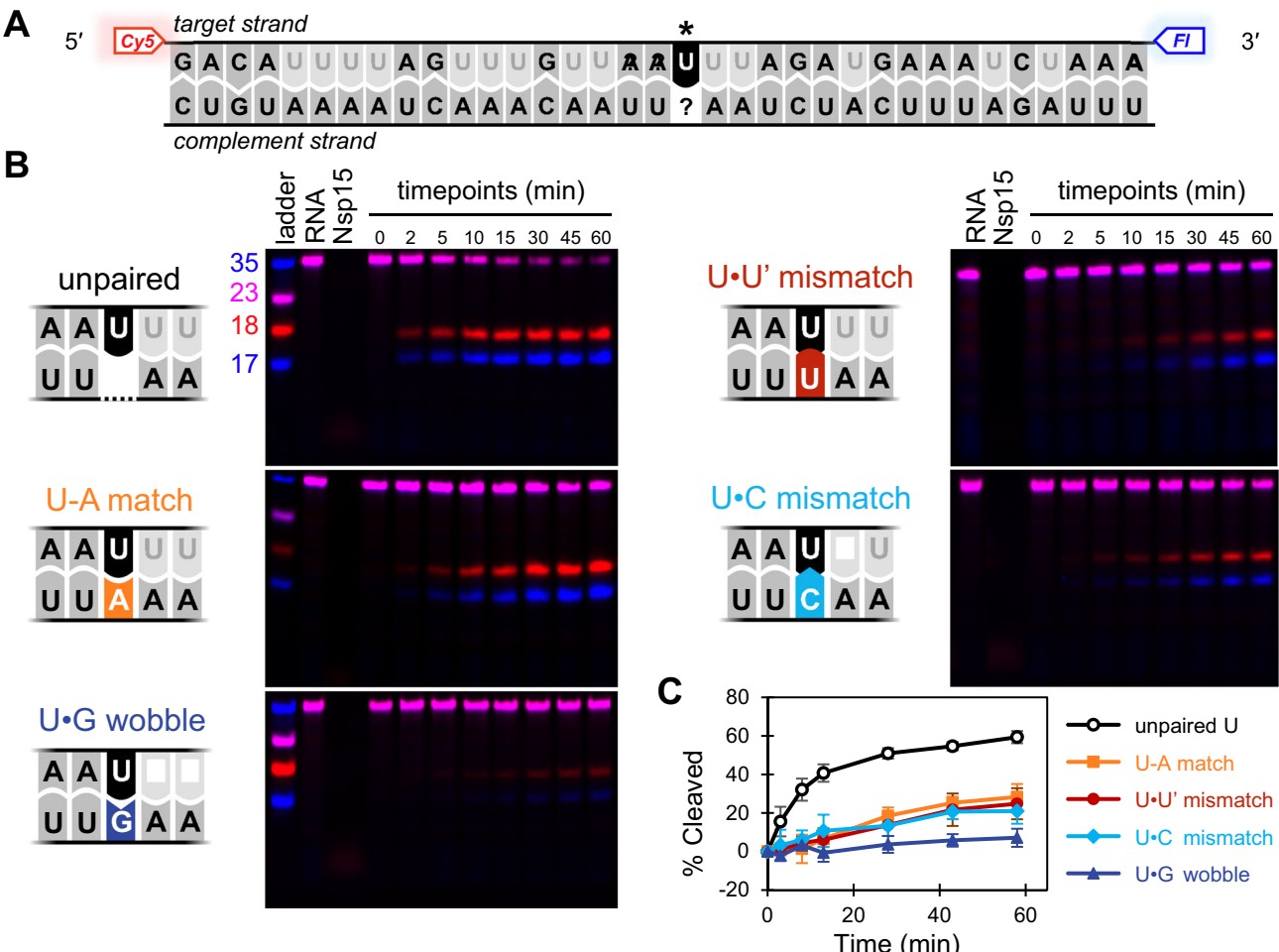

**Fig. 1 | U's complement affects its susceptibility to cleavage by Nsp15.** Unpaired U is especially susceptible, while U•G is especially resistant. **A** dsRNA oligo substrate design. The target strand, containing U19 (marked with *) is labeled with 5′ Cy5 and 3′ Fl. Us in the target strand other than U19 have been substituted for 2′-F-U (light gray) which is not cleavable by Nsp15. The complement strand is unlabeled. Five substrates were generated with this design, changing only the nucleotide complementary to U19 (unpaired U, U-A, U•G, U•U', or U•C). **B** Representative denaturing PAGE gels of timecourse nuclease assays for each of the five substrates, performed at 50 nM Nsp15 and 500 nM dsRNA. Uncleaved RNA appears in both Cy5 (red) and Fl (blue) channels, resulting in a pink signal in the overlaid image. **C** Percent of target strand cleaved over time, quantified via the intensity of the uncleaved RNA band and normalized to the 2 min timepoint. Each point with error bars represents average and standard deviation for three independent reactions, each using a distinct protein preparation (*N* = 3 biological replicates). Images of each gel and quantification data are provided in the Source Data file.

U•G is significantly less than cleavage of all other substrates except for U•C (*p* = 0.0000, 0.0065, 0.0311, 0.1220 for U•G versus unpaired U, U-A, U•U', U•C respectively). The exceptional resistance of the U•G to cleavage demonstrates that our substrate design is sufficient to focus cleavage on our target U with minimal effects from potential cleavage events in the complement strand. We note that in these assays, dsRNA cleavage plateaus over the course of the hour for all substrates, even those where total cleavage remains low. We attribute this to the presence of 2′-F-Us in the target strand, which continue to be sampled by Nsp15's active site (potentially at a greater frequency than the target U) but are not able to undergo cleavage.

We also tested susceptibility to cleavage for an alternate U•U mismatch, where both the target U19 and its complementary U are cleavable (Supplementary Fig. 1). We found that the target U19 for this substrate was slightly more susceptible to cleavage than the U•U' mismatch discussed above (37 ± 8% vs. 25 ± 5%), but not enough to be statistically significant. Nicking at the complementary U in the U•U mismatch may contribute to the target U19's increased cleavability[12,24], though our assay does not have the statistical power to determine this with certainty. Increased thermodynamic stability of the 2′-F-Us may also affect the relative susceptibility to cleavage for U•U' vs U•U.

To determine if our substrate design impacted Nsp15's cleavage preferences, we then tested a new set of three dsRNA substrates containing either an unpaired U, U-A match, or U•C mismatch at a central position (U14) without any 2′-F-U restricting the location of cleavage (Fig. 2A). These substrates test the susceptibility of U14 to cleavage against ten other potentially observable U-A cleavage sites in the target strand. As before, the sequence was adapted from the TRS-N and the strand containing our target U14 was labeled at the 5′ and 3′ ends by Cy5 and Fluorescein dyes respectively. We elected to swap the labeled and unlabeled strands from our first experiment, in order to limit consecutive Us in the middle of the sequence and thus facilitate distinguishing cleavage at our new target U14. This also presented an opportunity to probe whether Nsp15 cleaves in any locations that would have influenced the reactivity of our first target.

In the U-A match control substrate, we observe a mixture of products from cleavage of at least five distinct sites. Of these, the most prominent products are the result of cleavage in the four positions closest to the 5′ and 3′ ends (i.e., blue bands nearest the top and bottom of the gel, Fig. 2B). This aligns with a previous study and supports that cleavage near dsRNA ends could be the result of breathing of the dsRNA[8]. Consistent with our 2′-F-U experiment described above,

cleavage of the U•C mismatch substrate is remarkably similar to that of the U-A match control. In both U-A and U•C substrates, Nsp15 primarily favors cleavage at the Us that are near the dsRNA ends.

In contrast, in the unpaired U substrate, Nsp15 demonstrates an exceptionally strong preference for cleavage at the unpaired U14 that almost fully outcompetes cleavage at all other sites. By 30 min of reaction time, the uncleaved target strand has almost completely disappeared (Fig. 2B)—nearly twice as fast as the pace of cleavage for U-A and U•C substrates, similar to our first experiment. It is also worth noting that, across all three substrates, some cleavage products are only observed after other cleavage events have taken place. For example, at later timepoints in the unpaired U substrate we see a decrease in the intensity of the U14 product bands and the appearance of new bands with smaller molecular weight, which result from subsequent digestion of the U14 cleavage products. Since bases near the ends of dsRNA appear to be more susceptible to cleavage than those that are internal, each cleavage event could enhance the susceptibility of nearby sites to subsequent cleavage. Our observation of this effect appears limited to two consecutive cleavage events (i.e., one event facilitates a second event), though the chain reaction nature of this effect may differ in vivo depending on reaction conditions. Both sets of cleavage experiments support that Nsp15 has a strong preference for Us with enhanced solvent accessibility – especially unpaired Us, but also Us near the ends of dsRNA.

## Sequence-defined model of Nsp15-bound dsRNA via cryo-EM and unpaired U

We hypothesized that the exceptionally strong preference Nsp15 demonstrates for unpaired U would be advantageous for cryo-EM and could provide insights into the structural features that predispose a dsRNA substrate to cleavage by Nsp15. Several cryo-EM reconstructions of Nsp15 bound to dsRNA have recently been published[8,44], clearly showing a flipped U engaged in Nsp15's active site. However, these structures are potentially limited by the presence of multiple equivalent U targets in the dsRNA substrate, which introduces heterogeneity in the dsRNA density outside of the flipped U. This heterogeneity can be difficult to resolve via data processing alone, potentially resulting in the averaging of local structural features around multiple Us into a single position in the map. By utilizing a dsRNA substrate containing one highly preferred unpaired U (same as in Fig. 2) and a catalytically inactive Nsp15 mutant (H235A), we were able to circumvent these challenges and build a sequence-defined model of Nsp15-bound dsRNA via cryo-EM (Fig. 3A–C, Supplementary Fig. 2). Our model helps to clarify that Nsp15's structure prioritizes non-base specific interactions with dsRNA, suggesting Nsp15 recognizes more general features of dsRNA targets such as shape, flexibility, and/or hydrophobicity rather than strictly searching for particular sequences.

Our final map has a resolution of 3.24 Å (Table 1), with many regions reaching a local resolution of 3.0 Å (Fig. 3D, Supplementary Fig. 3). The five unoccupied endoU domains of Nsp15 show lower local resolution (3.4–4.0 Å) than the middle and N-terminal domains (3.0–3.3 Å), particularly on the active site side of the apo endoU domains. The endoU domain of P1, which is engaging the dsRNA, shows higher local resolution than the five unoccupied endoU domains, in agreement with earlier structures showing that the endoU domain is more dynamic in the absence of RNA[33,45]. The cryo-EM density for the dsRNA shows more variable local resolution than that of Nsp15, ranging from 3.0 Å at the flipped-out U to 5.0–5.5 Å.

Interestingly, the dsRNA shows patches of high local resolution centered around areas where the RNA contacts Nsp15 (Fig. 3E), suggesting that the otherwise fairly flexible dsRNA is more constrained at these contact points. Nsp15 makes close contact with the dsRNA via a number of amino acid residues in the endoU domain of protomer P1, with a few supporting contacts in protomers P2 and P4, in agreement

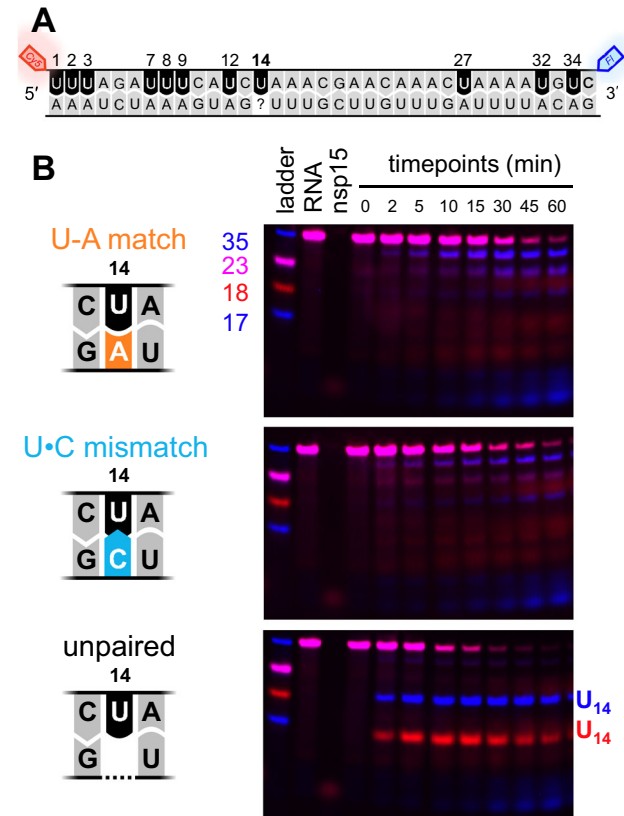

**Fig. 2 | Nsp15 prefers unpaired U over paired and mismatched Us, even in substrates without 2'-F modification. A** Substrate design. The target strand is labeled with 5' Cy5 and 3' Fl, and contains 11 cleavable Us. The complement strand is unlabeled. Three substrates were created with this design by changing the nucleotide complement to U14 to produce either a U-A, U•C, or unpaired U. **B** Representative denaturing PAGE gels of time-course nuclease assays for each of the three substrates. The U-A and U•C substrates show similar patterns of cleavage with Us in the 4 positions nearest the ends being most favored. The unpaired U substrate shows a different pattern, with cleavage at the unpaired U14 outpacing cleavage at all other Us. Note the near-complete disappearance of the target strand by 30 min, approximately twice as fast as in U-A and U•C substrates. Three independent reactions (each using a distinct protein preparation) were performed with each substrate. Images of each gel are provided in the Source Data file.

with previous structures that show dsRNA engages with a particular groove formed by three protomers of Nsp15[8,44]. (Fig. 3A shows a simplified map of locations in the dsRNA that are contacted by Nsp15; Supplementary Fig. 3D shows interactions between Nsp15 sidechains and the dsRNA in our atomic model; Supplementary Fig. 4 compares published atomic models of Nsp15 with dsRNA.) These interactions occur primarily in the minor groove of the dsRNA.

Examining the contact surface areas between Nsp15 and our dsRNA, we find that Nsp15 uses a variety of intermolecular interactions to hold its substrate, in agreement with previously determined structures[8,9,11,45,46]. In the active site of protomer P1, K290 (one of three catalytic residues) and S294 (responsible for discriminating U from other nucleotides) are within H-bonding distance of the flipped U. H250 (another catalytic residue) is poised next to the 2'-OH of the flipped U. In our structure, the third catalytic residue H235 has been mutated to an alanine to prevent cleavage from proceeding. Y343 uses its aromatic ring to stabilize the flipped U while using its hydroxyl group to form a weak H-bond with the 2'OH of the residue 5' of the U (C13). W333 is intercalated into the dsRNA helix, stacking with the base 3' of the flipped U (A15) and the closest complimentary base (U-21), which is in agreement with previously published structures

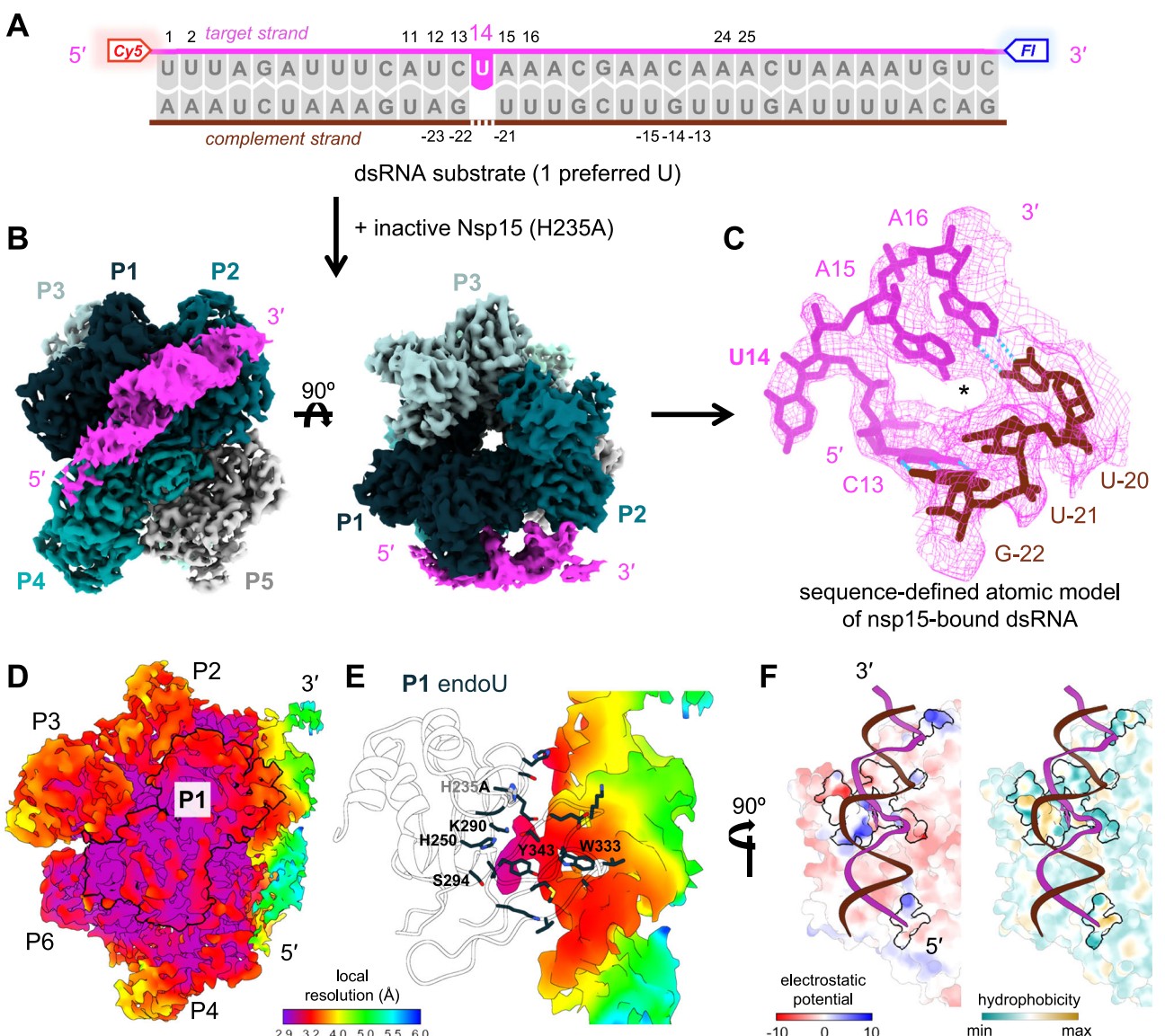

**Fig. 3 | Substrate containing highly favored unpaired U enables reconstruction of Nsp15-bound dsRNA with defined sequence. A** Design of dsRNA oligo incubated with Nsp15 for cryo-EM experiments; identical to unpaired U substrate used in Fig. 2. Nucleotides that were found to be sites of interaction with Nsp15 are labeled with their position number. Labels in the target strand containing the unpaired U are positive, with position 1 at the 5′ end; labels in the complement strand are negative, with position -1 at the complement strand's 5′ end. **B** Cryo-EM density map of dsRNA-bound Nsp15 (H235A mut) at a resolution of 3.24 Å, including dsRNA (pink), protomers of Nsp15 that interact with dsRNA (P1, P2, P4 in shades of dark teal), and remaining protomers of Nsp15 (P3, P5, P6 in shades of light blue-gray). **C** Closeup view of atomic model and mesh cryo-EM density map showing dsRNA residues surrounding the flipped-out U14, which is engaged in the active site of Nsp15 protomer P1. H-bonds are shown in cyan. **D** Cryo-EM density map of Nsp15-bound dsRNA, colored by local resolution. Protomer P1 is outlined in black. **E** Residues of Nsp15 protomer P1 calculated (via dr_sasa[47]) to interact with dsRNA, with P1 endoU domain represented as a transparent cartoon and dsRNA density map colored by local resolution. Residues previously shown to be key in Nsp15 cleavage of dsRNA are labeled; see Supplementary Information for an alternate version with all interacting endoU residues labeled. H235 (light gray) has been superimposed from a structure of apo WT nsp15 (PDB ID 7N33) over A235. **F** Binding pocket of Nsp15 for dsRNA, colored by electrostatic potential (left) and hydrophobicity (right, represented by the molecular lipophilicity potential) using default settings in ChimeraX. Residues calculated to interact with dsRNA are outlined in black and shown in full color according to key; all other residues of Nsp15 are included to provide structural context but shown at 65% transparency. See Supplementary Fig. 5 for alternate version with all residues of Nsp15 in full color.

(Supplementary Fig. 4)[8,44]. A handful of polar uncharged residues (S244, G248, S316, T341) appear close enough in our model to form H-bonds with the sugar-phosphate backbone of the RNA near the flipped U (at G-14, A15, U-21, and A15 respectively). Additional polar uncharged residues participate in stabilizing nonbonding interactions with the sugar-phosphate backbone across protomers P1 (Q245, G247), P3 (Q19, Q20, G147, S148), and P4 (T113, N137). These residues could potentially form H-bonds with dsRNA substrates in slightly different conformations than ours.

In addition to H-bonding and aromatic stacking interactions, we identified several positively charged (P1: H243, K335, K345; P3: K13, K65, K150; P4: K111) and hydrophobic (P1: V292, V315, M331, V318) residues that also interact with the dsRNA (Fig. 3F, Supplementary Fig. 3, 5) using the web-based software tool "dr_sasa."[47] (The "Contacts" tool in ChimeraX generates a similar set of residues.) These residues appear strategically placed to facilitate interactions between Nsp15 and dsRNA (Fig. 3F). While Nsp15 is overwhelmingly negatively charged, positively charged residues are concentrated around the

**Table 1 | Cryo-EM collection and processing statistics**

| Data collection and processing | Nsp15/dsRNA dataset 1 (0° Tilt) | Nsp15/dsRNA dataset 2 (30° Tilt) |
|---|---|---|
| Microscope | Titan Krios | Titan Krios |
| Detector | K3 | K3 |
| Nominal magnification | 130000 | 130000 |
| Voltage (kV) | 300 | 300 |
| Electron exposure (e⁻/Å²) | 50 | 50 |
| Defocus range (μm) | [-1.2, -2.2] | [-0.8, -1.8] |
| Pixel size (Å) | 0.67 | 0.67 |
| Symmetry imposed | C1 | C1 |
| Number of Micrographs | 7697 | 5509 |
| Initial particle images | 1,190,304 | 671, 713 |
| Final particle images | 217,900 (combined) | 217,900 (combined) |
| Refinement | Combined Datasets | |
| Resolution (Å)/FSC threshold | 3.24 | |
| B-factor used for map sharpening (Å²) | -144.2 | |
| *Map to model CC* | | |
| CC (mask) | 0.87 | |
| CC (volume) | 0.85 | |
| CC (peaks) | 0.79 | |
| CC (box) | 0.83 | |
| *Model composition* | | |
| Non-hydrogen atoms | 17576 | |
| Protein residues | 2084 | |
| Nucleic acid residues | 55 | |
| *Mean B factors (Å²)* | | |
| Protein | 73.72 | |
| Nucleic acid | 142.46 | |
| *R.m.s. deviations* | | |
| Bond lengths (Å) | 0.005 | |
| Bond angles (°) | 0.496 | |
| *Validation* | | |
| Molprobity score | 1.59 | |
| Clashscore | 4.55 | |
| Poor rotamers (%) | 1.36 | |
| *Ramachandran plot* | | |
| Favored (%) | 96.23 | |
| Allowed (%) | 3.77 | |
| Disallowed (%) | 0 | |

endoU domain to facilitate close contact with the dsRNA minor groove, and are also sprinkled near the 3′ and 5′ ends of our dsRNA (10–15 base pairs away from the flipped U). These positively charged residues are arranged in a V shape that parallels the kink we observe in our dsRNA substrate. Hydrophobic residues are clustered around the edge of the endoU domain, forming a plug that shields the hydrophobic core of the dsRNA around the flipped base from solvent.

Our sequence-defined atomic model shows that most interactions between Nsp15 and the dsRNA are mediated by the phosphate backbone and are not base-specific. Our model does suggest two potential ways in which dsRNA sequence could influence susceptibility to cleavage by Nsp15. First, we observe discontinuous base stacking and a disruption of H-bonding in the base pair 3′ of the flipped U, which presumably help relieve the strain created by the 1 nt bulge unpaired U and imposed by base-flipping; the identity of these bases would influence the ease with which these kinds of distortions to the dsRNA occur[24]. Second, our model shows Nsp15 making close contacts with a few of the bases neighboring the flipped U (Fig. 3A, C; Supplementary Fig. 6, Supplementary Table 2), though these form primarily stacking and hydrophobic interactions. While the identity of these bases may influence the ease with which Nsp15 can position a particular dsRNA around its active site, these interactions are modular rather than strictly sequence-specific. Thus, we propose that a particular dsRNA's susceptibility to cleavage is more likely to be related to qualities of the dsRNA itself (availability of the U, ability of the helix to deform, etc.) rather than any particular base-specific interactions between the dsRNA and Nsp15.

## Us with more solvent-accessible 2′ positions are more readily cleaved by Nsp15

Next, we investigated the influence of a U's neighbor context on its susceptibility to cleavage through nuclease assays. In parallel, we also probed the chemical environment of U in different neighbor contexts in the absence of Nsp15 via 1D ¹⁹F NMR[48,49]. We hypothesized that if Nsp15 is opportunistic—looking for a spontaneously flipped U rather than actively instigating base-flipping—we might see increased cleavage efficiency for Us that tend to exist in particular chemical environments (i.e., conformations).

To perform these experiments, we designed a new set of dsRNA oligos, again using 2′-F-Us to control which U is observable (Fig. 4A, black) and which Us are invisible (Fig. 4A, gray). For both experiments, we selected a single observable U at position 19 of a 35 nt duplex, and altered the two neighboring nucleotides on both the 5′ and 3′ sides of the U to one of five sequences (Fig. 4B); for each of these sequences, we tested both unpaired U and paired U·A. (Note that we avoided including sequences where U19 was flanked by C due to the possibility that the unpaired U could form a wobble pair with a nearby G in the complementary strand.) Cleavage assays were performed as described for Fig. 1. To perform our 1D ¹⁹F NMR experiments, we used a single 2′-F-U at position 19 as our observable U, with all other Us left as 2′OH (Supplementary Table 3). In these spectra, peak position reports information about the chemical environment(s) of the ¹⁹F label (i.e., the minor groove side of the U's sugar), and peak width reports information about the coexistence of multiple conformations[49,50]. This technique has previously been used to probe enzymatic capture of flipped bases in DNA[48], as well as interconversion between two conformations in a bistable RNA hairpin[49], but has not been used to probe spontaneous base flipping in dsRNA.

From our cleavage assays (Fig. 4C), we find that in all neighbor contexts, unpaired U is cleaved more readily than paired U·A, consistent with our earlier results. We also find that, secondarily, neighbor context affects the susceptibility to cleavage for both unpaired U and paired U·A. This effect is not strictly dependent on A/U content (see differences between sequences 1 "AAUUU" and 5 "AAUAA," or the similarity between sequences 2 "CGUGC" and 3 "AAUGC"). Also, sequence context does not uniformly have the same effects on unpaired U vs. paired U·A substrates. For example, in sequences 1 and 2, the unpaired Us are similarly susceptible to cleavage, but the U·A pairs are not. Sequence 4 has both the least-susceptible unpaired U and the most-susceptible paired U·A of the five sequences characterized.

From our 1D ¹⁹F NMR spectra of dsRNA in the absence of Nsp15 (Fig. 4D, Supplementary Table 4), we observe that for each sequence, the peak for the unpaired U is consistently shifted downfield (to the left) relative to the corresponding U·A signal. Though the identity of neighboring nucleotides affects the exact peak location, the range of shifts for unpaired Us does not overlap with those for paired U·As in this system. Previous literature suggests that base flipping events for paired bases are relatively rare and transitory[22,31,48,51], while depending on the sequence, unpaired bases can exist in metastable flipped[52] or stacked conformations[25], or interconvert readily between flipped and stacked states[53]. We therefore hypothesize that the downfield shift of

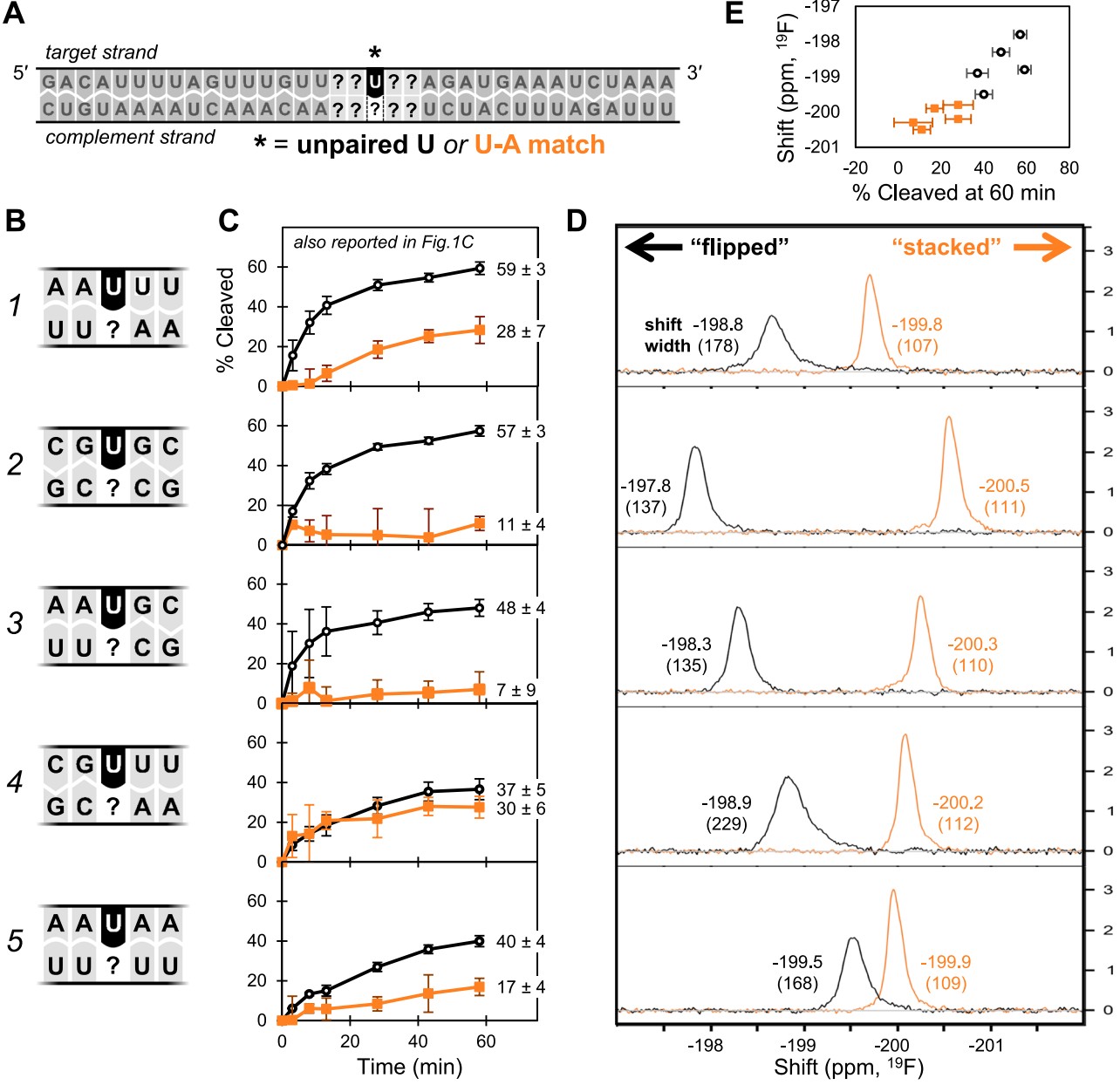

**Fig. 4 | A U's susceptibility to cleavage by Nsp15 parallels its tendency to spontaneously flip. A** dsRNA substrate design, with observable target U* (in black). For cleavage assays, U* = 2'-OH, non-target Us in target strand = 2'-F-U. Target strand has 5' Cy5 and 3' Fl labels, as in Fig. 1. For NMR, U* = 2'-F-U, all other Us = 2'-OH. From this design, a set of substrates were created by changing 2 nt on both 5' and 3' sides of U* (gray, marked with?) to different sequences, shown in (**B**). For each sequence, U* was either unpaired or engaged in a U•A pair (orange). **C** Percent of target strand cleaved over time, quantified via intensity of the uncleaved RNA band and normalized to the 2 min timepoint. Each point with error bars represents average and standard deviation for at least three independent reactions, each using

a distinct protein preparation (*N* = 3 biological replicates). Images of each gel and quantification data are provided in the Source Data file. Data for substrate 1 ("AAUUU") in panel C is reproduced from Fig. 1C for clarity and ease of comparison. **D** 1D $^{19}$F NMR spectra for each dsRNA substrate in the absence of Nsp15. Each peak is labeled with its shift (ppm, top value) and linewidth (Hz, bottom value, in parentheses). **E** Scatter plot of shift (peak position) from 1D $^{19}$F NMR spectra vs. % cleaved at 60 min from our enzymatic assays. For each point, x position represents the average of three independent reactions for a single substrate (error bars represent standard deviation), y position represents one spectrum.

our unpaired Us relative to their corresponding paired U-A signals is at least partially due to exchange with a flipped conformation. In other words, unpaired Us have greater "flipped character" (i.e., spend more time in a flipped or partially flipped conformation) than paired U-As.

Sequences 2 and 5 in particular ("CGUGC" and "AAUAA," respectively) show an interesting contrast in $^{19}$F NMR: sequence 2 has both the most extremely downfield unpaired U *and* upfield U-A of all sequences we studied, while sequence 5 has unpaired U and paired U-A with the most similar and intermediate shifts. This contrast highlights a trend that is true across all of our sequences – the effect of neighbor

context on unpaired Us opposes that on paired U-As. We propose that this contrast may be related to the conformational rigidity of each dsRNA oligo and the resulting ability or inability to accommodate a U in stacking interactions. A more rigid sequence might cause an unpaired U to be locked in a flipped conformation and a paired U-A to be locked in a stacked conformation (e.g., resulting in the strong difference in shifts seen in sequence 2). Likewise, a less rigid sequence might better accommodate the helix distortions required for an unpaired U to occasionally be stacked *and also* allow more frequent base flipping from a paired U-A (e.g., resulting in the intermediate

shifts seen in sequence 5). We suspect sequence 5 has lower rigidity around the U than sequence 2 due to its lack of G/C neighbors (fewer H-bonds between each neighboring base pair) and/ or the "UUUU" in its complementary strand that would better tolerate discontinuous base stacking[24].

We also observe that unpaired Us across all sequences have broader peak widths than the paired U-A substrates, indicating greater local conformational flexibility in the unpaired Us (Fig. 4D, Supplementary Table 4, Supplementary Fig. 7). Substrates with the unpaired U amidst consecutive Us (sequences 1 and 4) have the broadest peak widths of all, likely reflecting the coexistence of several subpopulations – for example, where our observable 2′-F-U is sometimes paired to a complementary A and one of its neighbors is unpaired, or the three consecutive Us are engaged in non-Watson Crick base pairing schemes with the two complementary As[54].

Comparing our cleavage assays and NMR experiments, we find a strong relationship between the strength of a U's flipped character and the U's susceptibility to cleavage by Nsp15. Percent cleaved at 60 min is positively correlated with peak shift, with a Pearson's r value of 0.89: substrates with downfield shifts in $^{19}$F NMR are more readily cleaved by Nsp15 (Fig. 4E). This relationship holds if we expand our dataset to include mismatched substrates (Supplementary Fig. 7, 8E), with the notable exception of U•G wobble pairs. The U•G wobble pairs exhibit strong downfield shifts relative to the paired and mismatched substrates, but are exceptionally resistant to cleavage by Nsp15. U•G wobble pairs (in the cis Watson/Watson configuration[32,54] used here) form H-bonds via different functional groups than Watson Crick U-A pairs; this changes the bond angle between the U/G bases and their sugar rings, tucking the U farther into the major groove and forcing the G to stick out slightly into the minor groove[55] (Supplementary Fig. 8F, G). Thus, we believe the downfield shift observed in our U•G substrates is the combined result of unusual proximity between the 2′-F of the U and the $NH_2$ of the G, as well as the change in glycosidic angle adjacent to the 2′-F that twists the U into the major groove. U•G wobble pairs are not stronger than U-A pairs[55], so we hypothesize that their resistance to cleavage by Nsp15 is related to geometric constraints that hinder favorable interactions between Nsp15 and the U•G (see Discussion).

## dsRNA secondary structure drives Nsp15 to select particular unpaired Us

Finally, we selected a dsRNA substrate that would allow us to enforce competition between Us in matched, mismatched, wobble, and unpaired contexts and simultaneously assess their relative susceptibility to cleavage. Specifically, we selected a portion of the highly structured 5′ untranslated region (UTR) of the SARS-CoV-2 genome called stem loop 4 (SL4), which has roles in regulating sub-genomic RNA synthesis during the viral replication cycle[52,56] and could be a physiological target of Nsp15 in vivo[16]. SL4 contains several U-A and U•G pairs, three unpaired Us (1 of which is in a one-nucleotide bulge and 2 of which are in the flexible loop), and one U•C mismatch. The hairpin structure of SL4 has been characterized by a number of techniques, including SHAPE[56,57], computational[56], and NMR[52] analyses, providing a rich structural understanding against which we can compare each U's susceptibility to cleavage by Nsp15.

First, we characterized different Us in SL4 by 1D $^{19}$F NMR, to better understand the effect of structure on $^{19}$F NMR shift. To accomplish this, we generated five derivatives of SL4, each with one U labeled with 2′-F (Fig. 5A). Based on the ensemble structures of SL4 solved by Vögele et al. (Fig. 5B)[52], we labeled an unpaired U which is primarily flipped (U95), an unpaired U which is often stacked with its neighbors (U104), a U-A pair (U115), a U•C mismatch (U112), and a U•G wobble pair (U87) for our $^{19}$F analysis.

The 1D $^{19}$F NMR spectra of selected Us in SL4 (Fig. 5C) follow the same trends we observed previously in our dsRNA oligos (i.e., in Fig. 4).

The U-A pair and U•C mismatch show narrow peaks that are shifted upfield (right) relative to the unpaired Us. The unpaired U that is primarily flipped (U95) shows a peak that is shifted strongly downfield (left), while the other unpaired U that is primarily stacked (U104) shows a shift that is remarkably similar to that of the U-A pair. We also note that this unpaired-but-stacked U, which adopts a variety of conformations in the ensemble structures by Vögele et al.[52], shows the broadest peak of the five Us we labeled. Lastly, the U•G wobble pair is shifted farther downfield than the U-A pair. This validates our hypotheses about the relationships between peak shift and flipped character, peak width and conformational heterogeneity, and bond torsion and peak shift.

Next, we characterized susceptibility of SL4 to cleavage by Nsp15 by generating another SL4 derivative: we appended Cy5 to the 5′ end, and an $A_4$-Fl tail to the 3′ end of SL4 (Fig. 5D). We performed cleavage assays followed by PAGE as described earlier, and confirmed the identity of cleavage products via a parallel experiment with quench conditions optimized for mass spectrometry (MS). We also performed cleavage assays followed by PAGE and by MS analyses on a hairpin with a slightly modified sequence (Us at positions 104 and 108 changed to As) to facilitate the assignment of cleavage sites (Supplementary Fig. 9).

From these cleavage assays, we find that Nsp15 strongly prefers the unpaired U in the 1 nt bulge to all other Us in SL4. Products that result from cleavage at this U (U95) dominate in both PAGE (Fig. 5E) and MS (Supplementary Table 5, Supplementary Figs. 9, 10). This U is clearly flipped almost fully out of the hairpin's helix in all but one of the ten consensus NMR structures published by Vögele et al.[52], was reported to have intermediate SHAPE reactivity by Rangan et al. demonstrating accessibility of its 2′-OH[56], and showed a strong downfield shift in our $^{19}$F NMR experiments. Secondarily, we see products resulting from cleavage at the two unpaired Us in the loop (U104 and U108) and the U•C mismatch (U112) at approximately equivalent intensities by PAGE—each lower than U95. Products corresponding to these four cleavage events (U95, 104, 108, and 112) appear simultaneously in early timepoints. In all cases, we can identify via PAGE both the 5′ and the 3′ products, demonstrating that these early cleavage events compete to be the first to take place on the otherwise uncleaved substrate. Notably, $^{19}$F NMR shift is not a perfect predictor of susceptibility to cleavage by Nsp15, with U112 being cleaved more and U87 less than would be expected based purely on shift. We emphasize the importance of controlling for sequence and secondary structure context when making comparisons between $^{19}$F shifts, as $^{19}$F shifts are sensitive to multiple confounding stimuli that can challenge interpretation. U112, for example, is adjacent to a U•G wobble pair that may affect bond torsions in dimensions not experienced by any other U we probed.

Interestingly, we also observe a small amount of product resulting from cleavage at C100, which appears more slowly than other bands. MS confirms the presence of the 5′-Cy5 product, which includes C100 as the 3′ terminal nucleotide. By denaturing PAGE, we observe the 5′-Cy5 labeled product increase at a disproportional rate to the corresponding 3′ -Fl labeled product (Fig. 5, Supplementary Fig. 9), suggesting this site becomes more reactive to Nsp15 after other cleavage events have occurred. We also observe a small amount of the 3′-Fl product of cleavage at U120 (a U•G wobble pair adjacent to a C•C mismatch) by both PAGE and MS (Fig. 5, Supplementary Table 5), though we do not observe the corresponding 5′-Cy5 product. This chain reaction of enhanced reactivity is in line with our observations that Us with greater solvent accessibility are more reactive to Nsp15 – each cleavage event introduces nicks that increase the flexibility of the dsRNA[24], making a new set of bases more accessible to Nsp15. We suspect both of these secondary cleavage sites (C100 and U120) were singled out by Nsp15 after SL4 had been nicked elsewhere because of greater than average flexibility at these bases: C100 is complementary

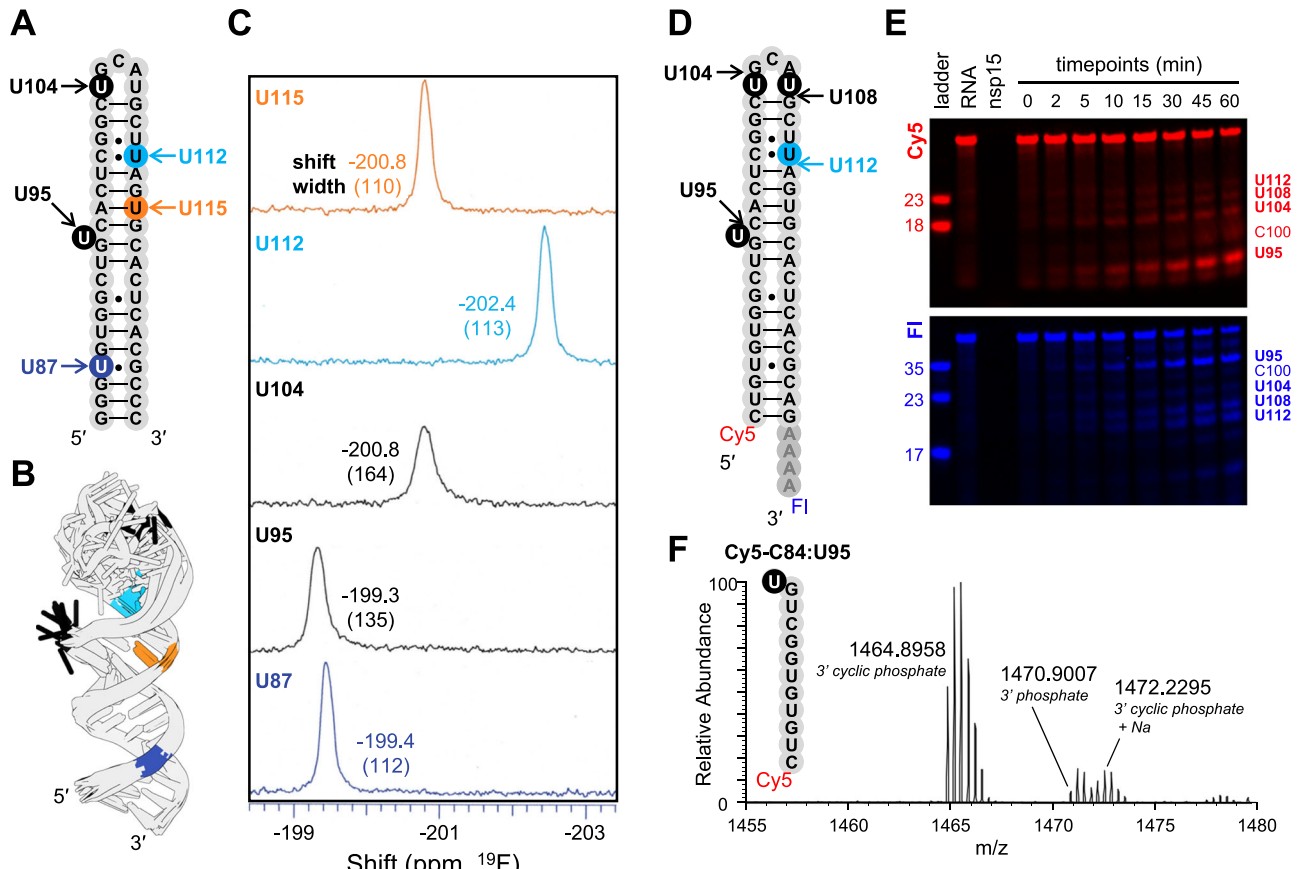

**Fig. 5 | Nsp15 prefers to cleave unpaired U in 1 nt bulge in stem loop RNA.**
**A** Design of substrates with sequence from the SARS-CoV-2 genome's Stem Loop 4 (SL4) used for ¹⁹F NMR. Key positions are marked where U was substituted for 2′-F-U to enable ¹⁹F NMR. Nts are labeled according to position in the positive strand viral genomic RNA. **B** Overlay of top ten structures of SL4 determined by Vögele et al.[52], with key Us color coded (PDBID 8CQ1). Note that U95 is consistently flipped out, while U104 and other Us are consistently stacked. **C** 1D ¹⁹F NMR spectra of five different SL4 derivatives, each with a single 2′-F-U at the indicated position. Each peak is labeled with its shift (top value) and linewidth (bottom value, in parentheses). **D** Design of SL4 substrate used for enzyme assay, with 5′ Cy5 and 3′ A₄-Fl. **E** Representative denaturing PAGE gel of time-course nuclease assays, with Cy5 (red) and Fl (blue) channels separated to facilitate band identification. Three independent reactions (each using a distinct protein preparation) were performed with each substrate. Images of each gel are provided in the Source Data file. **F** Mass spectrum showing multiply charged ions representing the major product of cleavage of SL4 by Nsp15 (at U95), corresponding to Cy5-C84:U95.

to one of the primary cut sites (U112), and U120 is adjacent to a C•C mismatch and only 3 base pairs away from the most favored primary cut site (U95).

## Discussion

Across multiple dsRNA substrate designs, we find that SARS-CoV-2 Nsp15's cleavage efficiency in dsRNA is driven by accessibility of the U. Nsp15 can cleave Us in a range of complement, neighbor, and secondary structure contexts, but Nsp15 cleaves unpaired Us most efficiently (Figs. 1, 2, 4, 5). We also find that % cleavage of a U by Nsp15 (under specific assay conditions) is correlated with the peak shift of that U (when labeled with 2′-F) in ¹⁹F NMR, which is related to the solvent accessibility of the 2′ position[31,48,49]. DsRNA substrates where a paired U-A is neighbored by Us, which pose a lower energetic barrier to base unstacking[19,20], are cleaved more readily than a paired U-A neighbored by As or Gs (Fig. 4). Further, Nsp15 strongly prefers to cleave the unpaired U95 in SL4 over all other Us, which in the absence of Nsp15 is fully extruded from the helix and solvent-accessible (Fig. 5)[52,56]. Base stacking influences the availability of other unpaired Us in SL4 (i.e., U104) and thus their reactivity to Nsp15.

Nsp15 shares some features with adenosine deaminases that act on RNA (ADARs), which rely on base flipping to access individual adenosine bases in dsRNA. ADAR interacts with the minor groove side of dsRNA, using a combination of H-bonding, electrostatic, and

hydrophobic interactions to contact the helix and the flipped adenosine[22,38,58]. ADAR projects residues into the helix to stabilize the flipped-out conformation, though unlike Nsp15's aromatic W333, ADAR uses residues that can H-bond directly with bases in the helix near its target (specifically, a glutamic acid flanked by glycines). These residues best accommodate certain nucleotides, partially driving ADAR's enzymatic preferences for targets in certain sequence contexts[22,38,58]. Nsp15 does not use the same context-specific scheme to interact with dsRNA, but interestingly, ADAR's enzymatic efficiency at a certain site is also driven by the target base's susceptibility to spontaneous base flipping–distinguishing targets that appear to be in otherwise equally favorable contexts[22,40].

Nsp15 also shares several features with enzymes involved in DNA lesion recognition, which address single-nucleotide lesions (misincorporations, chemical damage, mismatches, or unpaired bases) in DNA in a sequence-nonspecific manner, particularly glycosylases. While glycosylases are structurally diverse and have a range of targets, these enzymes all require base flipping to perform their functions, and prefer to act on substrates that are flexible and/or contain discontinuous base stacking[59]. Like Nsp15, glycosylases have small active sites that select a particular base via sterics and H-bonds, as well as a hydrophobic "reading head" that fills the space in the DNA duplex left by the flipped base[60]. Uracil glycosylase has been shown to increase the lifetime of the flipped-out conformation of DNA without affecting the

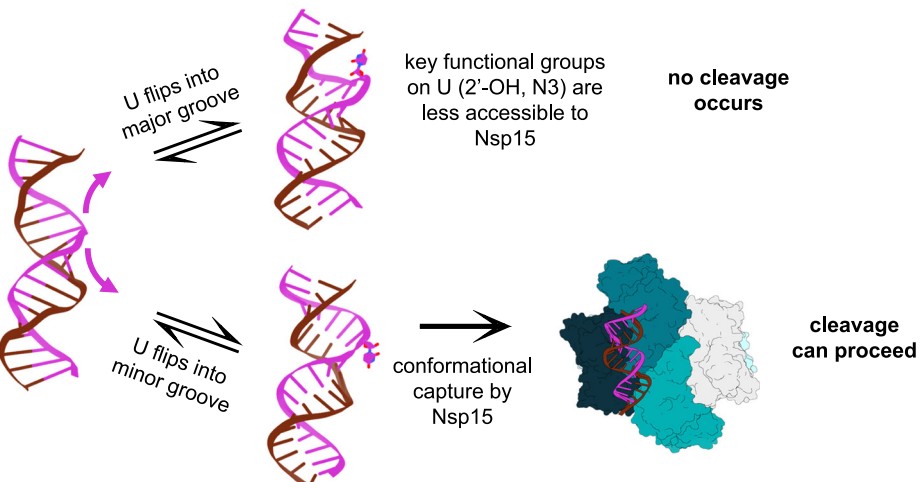

**Fig. 6 | Proposed mechanism for conformational capture of flipped U in dsRNA by Nsp15.** Bases can spontaneously flip towards either the major or minor groove of dsRNA. Nsp15 can only interact with key functional groups of the U if the U has flipped into the minor groove; these interactions are hindered when the base flips into the major groove, preventing cleavage. RNA is modeled from PDBs 7JL3, 8CQ1, and our model 9BIH.

rate of flipping[61]. This suggests that uracil glycosylase passively searches for substrates by taking advantage of spontaneous base flipping dynamics, rather than probing each base pair for lesions; the glycosylase's "reading head" does not need to actively push or pull a base to extrude it from the helix, but rather can simply capture a flipped base and prevent it from returning to the helix[62]. Increasing the flexibility and/or disrupting base stacking in a helix increases the likelihood of spontaneous base flipping, in turn increasing the efficiency of the enzyme[51,59–61]. Trapping bases that have spontaneously flipped is significantly more efficient for the enzyme than probing each base pair for DNA lesions[50,62].

Based on these similarities we propose a "conformational capture" mechanism[63] for Nsp15 (Fig. 6), where Nsp15 intercepts dsRNA substrates with a U flipped at least partially toward the minor groove, and uses its endoU domain to prevent the U from returning to the helix. In this mechanism, the direction of spontaneous base flipping would influence Nsp15's ability to effectively bind both the flipped U and the dsRNA's minor groove. Bases can spontaneously flip to either the major or minor groove, with computational calculations suggesting the energetic requirements for these two pathways in typical base pairs is similar[21,22]. However, NMR imino exchange experiments have shown that U•G wobble pairs flip almost exclusively towards the major groove[64], providing a mechanism by which Nsp15's cleavage of U•G wobble pairs is hindered. Even perfectly complementary nucleotides do undergo spontaneous base flipping[31,32], explaining Nsp15's ability to cleave at paired U-A without actively initiating base flipping. Our conformational capture mechanism is supported by recent kinetic analyses by Kalia et al., published during revisions of this manuscript, that demonstrate that Nsp15's specificity is driven primarily by RNA structure[65]. Still, Nsp15 may not be entirely passive in its approach to flipped Us, likely inducing minor local changes in the RNA conformation. Indeed, the minor groove of the dsRNA at the position of the flipped U is widened in our cryo-EM structure relative to the corresponding area around the flipped U95 in the NMR ensemble structures of SL4[52], suggesting Nsp15 may be capable of prying a partially-flipped U and its helix into place.

Overall, we find that Nsp15 most efficiently cleaves dsRNA substrates with a U already flipped into a favorable position, particularly U in a single nucleotide bulge amidst an otherwise helical duplex. This finding helps to unify a multitude of previous characterizations of Nsp15's cleavage preferences. Bulges appear frequently in viral genomes[57,66,67] and commonly serve as sites that stimulate specific interactions with other RNA, ligands, and proteins[25]. The cleavage activity of Nsp15 during infection may be partially driven by bulged or otherwise relatively accessible Us that appear at strategic positions in the coronaviral genomic RNA; Ancar et al. highlighted the presence of putative Nsp15 cleavage sites in structured regions in the positive strand of genomic RNA, including the Orf1a/1b frameshifting element and several TRSs[16], that serve to regulate viral replication and translation. Hackbart et al. determined that the polyU tail of the negative strand may also be a physiological target of Nsp15, which would be an example of direct digestion of RNA that would otherwise be recognized by the host immune system[13]. The relative importance of positive and negative strands of coronavirus genomic RNA in activating immune signaling has not been determined, and the structure of the negative strand has not been characterized for any coronavirus. Ultimately, Nsp15's cleavage activity during infection will be influenced by its association with other proteins in the double membrane replication vesicle, which likely mediates its access to viral RNA and may additionally affect RNA structure. The relative abundance of bulged versus paired Us in target RNA would also influence Nsp15's activity in vivo, with the potential to funnel cleavage towards less preferred substrates under the circumstance that more preferred Us are dramatically outnumbered. (The magnitude of this effect would depend on the frequencies and lifetimes of base flipping events towards the minor groove for all base pairs present.) Overall, our findings underscore the importance of RNA structure and dynamics in regulating coronaviral infection.

## Methods

### Expression and purification of Nsp15 from E. coli

Wild-type (WT) and mutant Nsp15 constructs used in this study were described previously[45], with N-terminal 6x His tag, thrombin cleavage site, and full-length Nsp15 (derived from the original SARS-CoV-2 viral isolate, GenBank NC_045512.2, codon-optimized for *Escherichia coli* expression) inserted into the bacterial expression vector pET-14b (ampicillin resistance). Nsp15 was overexpressed in *E. coli* C41 (DE3) competent cells by culturing transformed cells in Terrific Broth supplemented with ampicillin (100 mg/L) at 37 °C to an optical density (600 nm) of 0.2 in 1 L cultures. Cultures were then cooled for 1 h on ice, followed by induction with 0.2 mM isopropyl β-D-1-thiogalactopyranoside (IPTG) at 16 °C overnight. Cells were harvested and stored at −80 °C until use, typically in pellets derived from 2 L of cell culture.

To purify Nsp15, cells were thawed, resuspended in lysis buffer (50 mM Tris pH 8.0, 500 mM NaCl, 5% glycerol, 5 mM β-mercaptoethanol, 5 mM imidazole) supplemented with cOmplete EDTA-free protease inhibitor tablets (Roche), and lysed by sonication. Cell lysate was clarified by centrifugation at $26,915 \times g$ for 50 min at 4 °C, then incubated with TALON Superflow metal affinity resin (1 mL per 1 L cell culture) for 45 min at 4 °C. Resin-bound protein was washed with 5 column volumes of lysis buffer before being eluted from the resin with lysis buffer supplemented with 250 mM imidazole. Crude His-Nsp15 was buffer-exchanged into low-salt thrombin cleavage buffer (50 mM Tris pH 8.0, 150 mM NaCl, 5% glycerol, 2 mM β-ME, 2 mM CaCl$_2$, and 5–10 U thrombin per mg crude His-Nsp15) for His-tag cleavage at room temperature for 4 h. Cleaved His/Nsp15 was then repassed over TALON resin and quenched with 1 mM phenylmethylsulfonyl fluoride (PMSF) before final purification by size exclusion chromatography (SEC) using a Superdex-200 Increase 10/300 GL column equilibrated in SEC buffer (20 mM HEPES pH 7.5, 150 mM NaCl, 5 mM MnCl2, 5 mM β-ME). SEC-purified hexameric Nsp15 was supplemented with 0.2 mM dithiothreitol (DTT) before being stored at 4 °C. Within 48 h of SEC purification, Nsp15 was used for experiments and/ or concentrated at least tenfold and stored as a 50% glycerol stock at −20 °C. Glycerol stocks of Nsp15 were used exclusively for replicates of enzymatic assays and were kept at −20 °C.

## Preparation of dsRNA for nuclease assays

Unless being actively pipetted or imaged, RNA samples were protected from light to preserve fluorescent dye labels. Single stranded RNA (ssRNA) oligos were ordered from Horizon Discovery/ Dharmacon with HPLC purification in the 2′-ACE protected form and deprotected before use by incubating with acidic deprotection buffer (100 mM acetic acid, adjusted to pH 3.8 with TEMED) at 60 °C for 30 min, followed by evaporating to dryness via SpeedVac at 45 °C for 2 h. ssRNA oligos were then resuspended in UltraPure DEPC-treated water to 1 mM (concentration checked by microvolume UV-Vis spectrophotometer). To anneal double stranded RNA (dsRNA), two ssRNA oligos were combined in a 1:1 ratio in annealing buffer (20 mM HEPES pH 7.5, 100 mM NaCl, 5 mM MnCl$_2$) such that the final concentration of dsRNA = 250 μM, heated on a heat block to 75 °C for 5 min, then allowed to cool slowly on the heat block over 2 h. Annealing efficiency was assessed by running a sample of each dsRNA oligo on a 20% native polyacrylamide gel in 0.5X TBE buffer (Tris, borate, EDTA). Oligos were stored at −20 °C until use. Supplementary Table 1 includes a list of all oligos used for nuclease assays in this study.

## Preparation of hairpin RNA (SL4, SL4-1, and all 2′-F-U SL4 derivatives)

SL4 and SL4-1 oligos were ordered from Horizon Discovery/ Dharmacon with HPLC purification in the 2′-ACE protected form and deprotected, dried, and resuspended to 1 mM as detailed above. To anneal the hairpins, each oligo was diluted to 11 μM in UltraPure DEPC-treated water, heated to 80 °C for 5 min, spiked with 10x annealing buffer (to 10 μM RNA, 20 mM HEPES pH 7.5, 100 mM NaCl, 5 mM MnCl$_2$) and allowed to cool on ice 10 min. The crude hairpins were then purified via 20% native polyacrylamide gel in 0.5x TBE followed by extraction into 1 mL buffer each (10 mM HEPES, 50 mM NaCl, 0.01 U/μL RNase Inhibitor) for 1.5 h. After filtering to remove gel pieces, hairpin solutions were concentrated at least tenfold using 3k Millipore cellulose centrifugal filters, then stored at −20 °C until use.

## Nuclease cleavage assay

All cleavage assays were performed in triplicate with distinct preparations of protein. To perform one reaction, Nsp15 (50 nM) was incubated with dsRNA (500 nM) at room temperature in assay buffer (20 mM HEPES pH 7.5, 150 mM NaCl, 5 mM MnCl$_2$, 5 mM DTT, and 1 U/ μL RNasin) for 60 min. These concentrations of enzyme and substrate were previously optimized[8,11] for observing differences in cleavage between substrates over the course of 1 hr. Aliquots were taken from this reaction at 2, 5, 10, 15, 30, 45, and 60 min and quenched in an equal volume of 2x urea loading buffer (8 M urea, 20 mM Tris pH 8.0, 1 mM EDTA pH 8.0, 20% glycerol). Control samples containing RNA only, Nsp15 only, and RNA + Nsp15 at 0 min of reaction were also prepared in a 1:1 mixture of assay buffer and 2x urea loading buffer to confirm the integrity/ purity of the RNA and Nsp15 at reaction start. Dye was omitted from all samples except the Nsp15-only control to prevent potential overlap between similarly-sized dye and cleavage products. Samples were run on 15% polyacrylamide TBE-Urea gels (Invitrogen) alongside an RNA ladder containing a mixture of double- and single-labeled RNA oligos of different lengths. Gels were then imaged using a Typhoon RGB imager (Amersham) in red and blue channels ($\lambda_{ex} = 635$ nm, $\lambda_{em} = 655$–685 nm; and $\lambda_{ex} = 488$ nm, $\lambda_{em} = 515$–535 nm). For substrates containing 2′-F-U where cleavage was focused on a single target U (Figs. 1, 4, and Supplementary Fig. 8), RNA cleavage was quantified for each reaction by measuring the decreasing intensity of the intact RNA band in both red and blue channels, performing a baseline correction by subtracting the intensity of an empty well, then normalizing to the baseline-corrected intensity of the t = 2 min time-point (setting this intensity equal to "0% cleaved"). Measurements of % cleavage were averaged across red and blue channels for at least three biological replicates. Because 2′-F-Us can be sampled by Nsp15's active site but not cleaved, the % cleavage represents a ratio of how often the target U was sampled (and cleaved) relative to how often other Us in the sequence were sampled. Quantifying decrease in uncleaved RNA intensity was determined to be the least noisy reporter for susceptibility to cleavage by Nsp15 for these assays (as opposed to quantifying increase in product signal), since cleavage products can undergo subsequent digestion at off-target sites, including sometimes C instead of U. For this reason, reactions with native oligos (containing no 2′-F-U, i.e., Figs. 2, 5, and Supplementary Fig. 9) were not quantified.

## Cryo-EM sample preparation, data collection, and processing

Within 24 h of SEC purification, catalytically-dead Nsp15 (H235A mutant) was diluted in a low-salt buffer (20 mM HEPES pH 7.5, 100 mM NaCl, 5 mM MnCl$_2$, 5 mM DTT) to 0.75 μM and incubated with the unpaired U dsRNA in Fig. 2 (50 μM) at 4 °C for 2 h. Grids were prepared by sputtering C-flat R1.2/1.3 (Protochips) with a 30 nm thick layer of gold on the grid bar side using a Leica EM ACE-600 sputterer, then cleaned immediately before use with a Tergeo EM plasma cleaner (Pie Scientific) in immersion mode with a power of 38 W for a period of 75 s. A Leica EM GP2 freezing robot with a sample application chamber held at 95% humidity and 20 °C equipped with Whatman Grade 40 filter paper was used to freeze grids: 4 μL of sample was applied to the grid followed by a 5 s wait time, 3 s backblot, and plunge freezing into liquid ethane. Grids were transferred to liquid nitrogen for storage.

Data collection was performed using a Titan Krios electron microscope at 300 keV with a K3 Bioquantum detector using SerialEM[68] v4.0 or newer. Two datasets were collected, one at 0° tilt and one at 30° tilt, according to parameters listed in Table 1. Beam-induced motion and drift were corrected using MotionCor2[69] through Scipion 3[70]. CryoSPARC v3[71] was used in all subsequent image processing. Patch CTF estimation was performed on the aligned dose-weighted images. Exposures were curated by selecting for CTF fit resolution <6 Å. The Topaz wrapper in cryoSPARC[72] was trained on a set of about 7,000 good Nsp15 particles, identified via Blob Picking on 500 micrographs acquired at 0° tilt followed by 2D Classification. Particles were then picked from both datasets via the trained Topaz model and curated via 2D Classification. Ab-Initio Reconstruction with 3 classes was performed on particles from good classes of the 0° tilt dataset. Heterogeneous Refinement using the 3 output classes of this

Ab-Initio Reconstruction was used to curate particles from the 30° tilt dataset. Particles from the best classes of the Ab-Initio (0° tilt) and Heterogeneous Refinement (30° tilt) were then combined. Global CTF Refinement, 3D Variability Analysis, and iterations of Homogeneous and Non-Uniform Refinements generated the final density map. Processing is summarized in Supplementary Fig. 2.

### Model building

A SARS-CoV-2 Nsp15 cryo-EM structure (PDBID 7TJ2) was used as a starting model and fit into the cryo-EM maps using COOT[73]. A combination of rigid body and real-space refinement in Phenix[74] as well as iterative rounds of building in COOT were used to improve the fit of the models. The map was scaled by 0.9659, determined by Kabsch-Umeyama[75] least-squares of the model to a 1.85 Å homologous reference structure (PDBID 6WXC). Refinement included restraints for Ramachandran geometry, rotamers, and secondary structure, and for hydrogen bonds for U in the active site. Molprobity[76] was used to evaluate the model (Table 1). Figures were prepared using ChimeraX[77].

### Preparation of dsRNA and data acquisition for 1D $^{19}$F NMR spectroscopy

Single stranded RNA oligos were ordered from GenScript with HPLC purification in the 2' deprotected form. Oligo sequences are listed in Supplementary Table 3. Oligos were annealed as detailed for enzymatic experiments, except: labeled and unlabeled strands were combined at a ratio of 1:1.007; annealing buffer contained 5 mM $MgCl_2$ instead of $MnCl_2$, as Mn decreases signal-to-noise in NMR experiments; and the final concentration of dsRNA = 125 μM with final volume = 250 μL. Prior to NMR acquisition, trifluoroacetic acid (TFA) and $^2H_2O$ were added so that the final sample was 100 μM TFA and 8% (v/v) $^2H_2O$ for shift referencing (−75.25 ppm) and frequency lock, respectively. 17,000 transients were acquired with a pulse flip angle of 30 degrees, 0.7 s acquisition and 1 s recycle delay on an Agilent DD2 console operating with a $^{19}$F frequency of 564 MHz at a temperature of 298 K. Spectra were analyzed with VNMRJ 22.1 (Agilent) and Chenomx 11 (Alberta, Canada) to fit the chemical shift and linewidth.

### Mass spectrometry

An enzyme assay was performed with SL4 and SL4-1 hairpin substrates as described above, except: the entire volume of reaction was halted at 35 min by flash-freezing in liquid nitrogen. Frozen samples were stored at −80 °C until being prepared for injection.

Analyses were performed similarly to those described by Huang et al.[78]. Briefly, HILIC mobile phase A (MPA) was composed of 70% ACN buffered with 15 mM ammonium acetate (pH 9.0−adjusted with ammonium hydroxide), and mobile phase B (MPB) was composed of 30% ACN buffered with 15 mM ammonium acetate (pH 9.0 − adjusted with ammonium hydroxide). LC-MS analyses were conducted using a Vanquish UPLC system coupled with a Q Exactive Plus mass spectrometer with a HESI source, all sourced from Thermo Fisher Scientific (Waltham, MA, USA). HILIC chromatography was performed using a BEH Amide UHPLC column (Waters, Milford, MA, USA; dimensions: 2.1 mm × 150 mm, particle size: 1.7 micrometers, pore size: 130 angstroms) at a flow rate of 0.25 mL/min, with a column temperature of 30 °C. Frozen samples were thawed on ice for 10 min, the injected directly (5 microliter injections) and eluted with a linear gradient, gradually increasing from 20 to 70% MPB over 10 min. The gradient was then stepped to 80% MPB over 1 min and maintained for 2 min before reverting to 20% MPA, followed by re-equilibration at 20% MPA for 10 min before the subsequent injections. Mass spectrometer settings included: negative polarity, 3.0 kV spray voltage, S-lense at 50 volts, sheath gas (40 a.u.), auxiliary gas (15 a.u.), sweep gas (0 a.u.), capillary temperature of 325 degrees Celsius, vaporizer temperature of 350 degrees Celsius, mass range 400−2000 m/z, mass resolution 70,000, automatic gain control (AGC) 5e5, and maximum injection time (IT) of 100 ms. Data were analyzed in the Qual Browser application in the Xcalibur software suite (Thermo Fisher Scientific, Waltham, MA, USA) by summing approximately 15 s retention time windows across the entire chromatogram and manually inspecting the resulting spectra for multiply charged species across the entire m/z range. Analyses using BioPharma Finder were done using the isotopically resolved Xtract function with sliding windows and limiting the time range from 4 to 18 min (the RNA oligo elution window). Additional settings included a Target Avg Spectrum Width of 0.1 min, a 25% Offset, an Output Mass Range of 2,000 to 20,000 Daltons, a S/N Threshold of 3.00, a Rel. Abundance Threshold of 0%, a Charge Range of 2−20, a Min. Num Detected Charge of 2, enablement of Negative Charge, and the Isotope Table set to Nucleotide.

### Reporting summary

Further information on research design is available in the Nature Portfolio Reporting Summary linked to this article.

## Data availability

The cryo-EM maps and atomic coordinates for Nsp15 have been deposited in the Electron Microscopy Data Bank and PDB under the following access numbers: EMDB-44590 [https://www.emdataresource.org/EMD-44590] and PDB ID 9BIH. Mass-spec data have been deposited at Massive under the accession code MSV000094614 [https://doi.org/10.25345/C52R3P765]. Datasets we have reused include structural datasets from the PDB 7N33, 7JL3, 8CQ1, 7TJ2 and 6WXC. Source data are provided with this paper.

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

## Acknowledgements
We would like to thank all the members of the Molecular Microscopy Consortium at the NIEHS for their help with cryo-EM data collection and processing, along with all of the members of the Mass Spectrometry Research and Support Group at the NIEHS for their help with mass spectrometry data collection and analysis. This work was supported by the US National Institutes of Health Intramural Research Program; US National Institute of Environmental Health Sciences (NIEHS) (ZIA ES103247 to R.E.S., 1ZI CES102488 to J.G.W., 1ZICES103362 to G.A.M., and ZIC ES103326 to M.J.B). This work was also supported by the NIH Intramural Targeted Anti-COVID-19(ITAC) Program funded by the National Institute of Allergy and Infectious Diseases [NIAID, NIAID/ NIH 1ZIAES103340 (RES)].

## Author contributions
Z.M.W. and R.E.S. designed experiments and prepared the manuscript, which was reviewed and edited by all the authors. Z.M.W. and M.N.F. designed RNA substrates. M.N.F optimized nuclease assay conditions and provided technical advice. Z.M.W. prepared protein and RNA samples for nuclease assays, cryo-EM, and NMR with assistance from I.M.W., and I.S.H. Z.M.W., K.J.B., J.M.K., and M.J.B. performed cryo-EM screening, data collection, 3D reconstruction, and model building. J.G.W. acquired and analyzed mass-spec data. G.A.M., E.F.D., and S.A.G. performed NMR experiments and data analysis.

## Funding

## Competing interests
The authors declare no competing interests.
