## [Peer review File · Nature Communications]

REVIEWER COMMENTS

Reviewer #1 (Remarks to the Author):

The manuscript by Wright et al. presents a biochemical, biophysical, and structural study for understanding the dsRNA substrate cleavage preference of Nsp15, an important endoribonuclease in regulating viral dsRNA levels and viral replication by cleaving the 3' of uridine. Building on their previous structural and functional work, the authors aim to address whether Nsp15 initiates the flipping of uridines or captures uridines that spontaneously flip within the dsRNA substrate. In the present study, the authors designed a series of synthetic fluorinated dsRNA substrates for Nsp15-dependent cleavage, which have unveiled uridine's susceptibility within different secondary-structural contexts. Using their biochemical findings to define a preferred substrate, the authors determined the cryo-EM structure of Nsp15 bound to a dsRNA containing an unmatched uridine. The authors further applied ^{19}F NMR to monitor the spontaneous base flipping of uridines in the absence of Nsp15 and observed a correlation between the ^{19}F chemical shifts in uridine and the cleavage efficiency for different dsRNA substrates. These results led the authors to propose a conformational capture mechanism for Nsp15 cleavage, where spontaneous base flipping of uridine influences the substrate preferences of Nsp15. Overall, this is a well-presented study that provides new insights into substrate recognition by Nsp15 and should be of interest to the readers of Nature Communications. The review has the following comments and suggestions for the authors to consider in a revised manuscript.

1. The authors suggested that the strong preference for an unpaired U provides an advantage for cryo-EM study of the Nsp15 and dsRNA complex, which could reveal new structural insights into the Nsp15 binding of the substrate. However, the authors need to clarify whether this new structure unveiled new interactions absent in the previous structures. It would be beneficial to include discussions and figures to compare the new and existing structures and highlight their new findings. These new structural features should then be followed by mutagenesis and assay to confirm their functional importance.

2. A significant emphasis of the present study is the base flipping within dsRNA substrates, where the authors used the ^{19}F chemical shift of uridines as an indicator for base flipping and stacking. However, the reviewer notes that this approach is too preliminary. Given chemical shifts are highly sensitive to local environments, variations in neighboring nucleotides can substantially affect chemical shifts. For example, the $2'\text{-F}$ chemical shift for Us in the A-U base pair varies by nearly one ppm among sequences studied, as shown in Figure 4. Furthermore, in an RNA sequence different from the ones in the present work, its $2'\text{-F}$ chemical shift of U is downfield shifted when base-paired relative to its unpaired state, opposite to the observations reported here. Hence, to reliably use chemical shifts as an indicator for flipped and stacked Us, the authors need to establish reference chemical shifts for both conformations for each sequence. One suggestion could be measuring

chemical shift as a function of temperature, where the unpaired U has a higher likelihood of flipping out at a higher temperature. The authors could also incorporate the Stem Loop 4 sequence, whose U95 is known to flip, to further validate their approach.

3. Based on the data from the Stem Loop 4, the authors suggested that cleavage by Nsp15 at one site within dsRNA can increase the subsequent reactivity of nearby bases. Since U95 is the preferred cleavage site for Nsp15, the increased cleavage at U120 following U95 cleavage could potentially be attributed to U120 becoming the only cleavage site available for Nsp15 rather than an actual increase in its intrinsic cleavage rate. Could the authors provide further elaboration on their proposed mechanism?

4. It would benefit the readers if the authors could quantify the cleavage assays presented in Figs 2 and 5, similar to the quantification provided in Figs 1 and 4. This will provide consistency in data analysis and presentation.

5. The authors discussed the linewidth of the ^{19}F NMR data in the manuscript. However, it was unclear where these data were specifically presented, presumably the values in parentheses in Figure 4. To improve the clarity, the authors need to reference these data properly and include detailed information in the figure legend.

Reviewer #2 (Remarks to the Author):

The manuscript addresses the question of whether the endonuclease Nsp15 targets U residues that are already flipped or whether it induces flipping. The authors combine *in vitro* assays, cryoEM, and NMR analysis of U conformation to provide the answer and reveal important principles of specificity for an important covid enzyme.

There is a fair amount of qualitative data in the literature identifying some of the basic characteristics of preferred substrates for Nsp15. However, quantitative data regarding alternative substrates that have been carefully designed to test specific features of RNA structure are very much lacking. Moreover, several studies have now shown that RNA structure is important for Nsp15 specificity. Both the protein and RNA structure are dynamic making this aspect another key issue to address for the function and specificity of this enzyme.

Therefore, this manuscript represents an important step forward since it builds on both of these areas where mechanistic detail is needed. The data establish new insights into structure specificity of Nsp15 and are sufficiently rigorous and quantitative to provide benchmarks for more detailed study of this class of enzymes.

In my view there are two important issues (1 and 2 below) that need to be resolved and three additional minor comments (3-5) to consider.

1. The authors need to include the substrate and enzyme concentrations in the text and figure legends. They need to introduce and discuss the rationale for selecting these concentrations of E and S and rationalize their results in terms of whether the way they set up their reactions will reflect differences between alternative sites due to catalysis, binding, cooperativity, or some other aspect of the reaction.

2. There is an apparent inconsistency between the data in Figure 4 and the model for Nsp15 recognition in Figure 6. Several substrates (e.g. #1, 4 and 50) do not react to completion (only 10-50%). This effect is independent of the rate based on the data shown in Figure 4. However, the mechanism shown in Figure 6 predicts that the entire population of RNA should be cleaved because the different structural forms of the substrate are in equilibrium. In this model all RNAs will necessarily sample the U flipped conformation and eventually become a substrate. Are the authors claiming that flipping is so slow that there is no exchange between RNA conformations over the time course of the reaction? Does this conclusion follow from the NMR data? Some additional explanation is needed to resolve this apparent conflict between the data and the model.

3. In Figure 1C, does the U-U mismatch substrate have a 2'-F-U modification on the non-labeled strand? This seems to follow from the overall description of the substrate design, but it would be useful for the authors to clarify this since it impacts the interpretation to some degree.

4. The conclusion that bulged U is a key determinant for Nsp15 cleavage is important. However, an excess of dsRNA binding sites will necessarily reduce the rate of cleavage at a small number of bulged sites. In the discussion the authors might mention that the concentrations of alternative substrates in addition to intrinsic cleavage rate also determines relative cleavage site preference *in vivo*.

5. There are some minor typos in lines 461 and 462

Reviewer #3 (Remarks to the Author):

Reviewer #4 (Remarks to the Author):

- What are the noteworthy results?

This really interesting paper by Wright and coauthors investigates the really important question on the mechanism of dsRNA cleavage of the SARS-CoV2 protein NSP15 (alternate name NendoU), which is a key enzyme responsible to “hiding” the presence of the virus from the host immune system. It is present in all coronaviruses. The authors use a very impressive method spectrum, including functional cleavage assays, Cryo-EM, NMR, and mass spectroscopy to investigate the hypothesis that base-flipping of the U substrate is essential for the cleavage mechanism. They designed different dsRNA substrates, where they systematically varied the base pairing of the “substrate U” as well as the surrounding bases and also investigated the cleavage preference of one of the potential native substrates, the Stem Loop 4 from the SARS-CoV-2 genome. They discovered that the Nsp15’s cleavage efficiency in dsRNA is driven by the accessibility of the U. They were able to show that Nsp15 cleaves unpaired Us most efficiently but can also cleave Us in a range of complement, neighbor, and secondary structure contexts. Here some initially unexpected results were obtained, for example, that cleavage still occurs, when the U is fully base paired with A, but cleavage is nearly completely abolished in the case of a U-G wobble pair. In the discussion, they presented a very convincing interpretation of this unexpected result: The catalytic site of Nsp15 will only be able to access and cleave base flipped U when it is flipped into the minor groove of the dsRNA, which is the preferential orientation in unpaired U and also fully base-paired U-A but in the case for the U-G wobble pair U is flipped in the major groove of the ds-RNA and can therefore not be cleaved by Nsp15. They also varied the neighboring bases of the U-A in the dsRNA substrates and discovered that there is a high cleavage efficiency when the paired U-A is neighbored by Us, which could be explained that the presence of neighboring Us leads to a lower energetic barrier to base unstacking compared to a paired U-A neighbored by As or Gs. The authors then investigated the conformation of the different substrates (without Nsp15) by 19F NMR where they found that % cleavage of a U by Nsp15 is correlated with the peak shift of 2'-F-U in 19F NMR, which is directly related to solvent accessibility of the 2'-F. The NMR data were correlated very well with the results on the cleavage as the peak shift is most pronounced in dsRNA with unpaired U. Finally, they were able to show that Nsp15 also strongly prefers to cleave the unpaired U95 in SL4 over all other Us, where previous studies have shown that U85 is fully extruded from the helix and

solvent-accessible. The cleavage essays and NMR experiments are further supported by a cryo-EM structure of Nsp15 at medium resolution, where they solved the structure of Nsp15 in complex with the same dsRNA with an unpaired U that was used for the functional studies shown in Figure 2. They identified the dsRNA in the binding groove of subunit P1 of the hexameric Nsp15 and proposed a network of amino acids that interact with the dsRNA and specifically the base flipped U in the minor groove of the dsRNA, which is slightly widened in the structure of the Nsp15-dsRNA complex.

In conclusion, this exciting paper shows that Nsp15 does not actively base-flip the U but binds to and efficiently cleaves dsRNA substrates where U is in a base-flipped conformation. The widening of the minor groove could be induced by Nsp15 and thereby could contribute to the stabilization of the base flipped conformation and thereby could explain the efficient cleavage of fully based paired U-A where U base flipping is only a transient event.

- Will the work be of significance to the field and related fields? How does it compare to the established literature? If the work is not original, please provide relevant references.

The work is of very high significance to the field as it presents a very important finding of the mechanism of the cleavage of dsRNA by Nsp15. The work presented here is original and the conclusions are supported by a very carefully selected spectrum of methods ranging from the systematic design of dsRNA substrates to functional cleavage assays which are combined with structural studies by Cryo-EM as well as NMR studies on the dsRNA substrates. The authors relate their work to the existing literature leading to a very convincing picture of the mechanisms of dsRNA cleavage by Nsp15. It was especially impressive that their studies on the stem-loop 4 showed that Nsp15 cleaves this substrate nearly exclusively at U95 site which has been shown to be fully base flipped in previous studies.

The results presented here and the conclusions are of extremely high relevance to understanding the cleavage mechanism and may also have a huge impact on the development of potential future drugs that could inhibit the Nsp15 function and thereby prevent SARS-CoV-2 (and future coronaviruses that will become threats for human health) to hide the virus from the immune system.

- Does the work support the conclusions and claims, or is additional evidence needed?

The work presented here fully supports the exciting conclusions about the mechanism of NendoU “preying on based flipped U presented in this paper.

- Are there any flaws in the data analysis, interpretation, and conclusions? Do these prohibit publication or require revision?

The analysis of the data is done very carefully and the reviewer has not identified any flaws in the data analysis and interpretation. I have some comments and suggestions cc the representation of the Cryo-EM data in the Figures listed below, which should be revised to show more details of the interaction of the dsRNA and especially the base flipped U with the amino acids in the catalytic site,

which should be shown not just as a stick model but showing also the experimental density map for both the protein and the dsRNA. This revision should be included in the revised version of the paper.

- Is the methodology sound? Does the work meet the expected standards in your field?

The methodology is solid. All experiments are carefully designed and executed. The reviewer also really appreciated that the authors tell a very convincing story in their paper where they walk the reader through the reasoning for each of the consecutive experiments conducted so that the reader can follow nicely the experimental plan and see how the results of one experiment informed the experimental plan for the next step.

- Is there enough detail provided in the methods for the work to be reproduced?

The methods section of the paper is very detailed and describes all experiments in sufficient detail including isolation of the Nsp15 WT and its non-active mutant (used for the Cryo-EM studies), which can bind the substrate but cannot cleave it.

Below you will find some more minor comments and suggestions that should be included in the revision of the manuscript:

Cryo-EM structure analysis and interpretation:

The authors show multiple similar overview figures of the structure of Nsp15 bound to the ds RNA substrate but only one figure (Figure 3 D) also shows the experimental density for the small 6 nucleotide section of the dsRNA that includes the flipped U14 base. While this density looks very convincing none of the figures show experimental densities for the protein. As the authors include a detailed discussion on the amino acids that interact with the dsRNA this should be supported by showing the density maps for key residues that foster the protein dsRNA interactions. These maps with the side chain densities should be included in the supplementary figures S2 and S3. It would be also helpful to include a table with the shortest edge-to-edge distances of key residues with the dsRNA in the supplementary information.

Further suggestions on Figures:

Supplementary Figure S4

Panels C and D: the light and dark blue colors are nearly indistinguishable. Please choose a different color scheme here.

Panel E: The authors have indicated that all experiments were performed in triplicates. Please include error bars in this figure derived from the triplicate experiments.

Figures 1 and 2 (gels) and the interpretation of the cleavage results

The authors have based their analysis of the % cleavage on the scanning of the intensity of the non-cleaved substrate, instead of the appearance of the cleavage products and the reviewers wonder why as the appearance of the cleavage products is much more prominently visible in all gels compared to the diminishing of the non-cleaved dsRNA. Looking at Figure 1 specifically there is a discrepancy between the time course shown in Figure 1C (based on the disappearance of the

dsRNA band) and the time course of the appearance of the cleaved products for the A-H fully base-paired sample and U-U mismatch. The interpretation of the results based on the dsRNA band shows an order of cleavage efficiency of U>U-U mismatch >U-A match>U-C mismatch >U-G wobble. However, when following the appearance of the cleaved bands the order is different, where the cleavage efficiency of the U-A match is second to the unpaired U. For the unpaired U cleavage products are clearly visible at 5 min while cleavage products for the U-A match already prominently appear already just 5 min later at 10 min while the cleavage bands for U-U and U-C do not appear before 15 and 30 minutes. This order where the U-A match is faster and more efficiently cleaved compared to the mismatches U-U and U-C is also visible from the intensity of the cleavage bands which are more prominent for U-A than for U-U and U-C. It would be important that the author explain why they do not use the cleavage products for the interpretation of the cleavage efficiency and why the order of cleavage efficiency is different when looking at the cleavage products.

Rebuttal NCOMMS-24-21724

Reviewer comments (black text) and our response (*blue italics*)

Reviewer #1 (Remarks to the Author):

The manuscript by Wright et al. presents a biochemical, biophysical, and structural study for understanding the dsRNA substrate cleavage preference of Nsp15, an important endoribonuclease in regulating viral dsRNA levels and viral replication by cleaving the 3' of uridine. Building on their previous structural and functional work, the authors aim to address whether Nsp15 initiates the flipping of uridines or captures uridines that spontaneously flip within the dsRNA substrate. In the present study, the authors designed a series of synthetic fluorinated dsRNA substrates for Nsp15-dependent cleavage, which have unveiled uridine's susceptibility within different secondary-structural contexts. Using their biochemical findings to define a preferred substrate, the authors determined the cryo-EM structure of Nsp15 bound to a dsRNA containing an unmatched uridine. The authors further applied ¹⁹F NMR to monitor the spontaneous base flipping of uridines in the absence of Nsp15 and observed a correlation between the ¹⁹F chemical shifts in uridine and the cleavage efficiency for different dsRNA substrates. These results led the authors to propose a conformational capture mechanism for Nsp15 cleavage, where spontaneous base flipping of uridine influences the substrate preferences of Nsp15. Overall, this is a well-presented study that provides new insights into substrate recognition by Nsp15 and should be of interest to the readers of Nature Communications. The review has the following comments and suggestions for the authors to consider in a revised manuscript.

We thank the reviewer for their supportive and constructive comments.

1. The authors suggested that the strong preference for an unpaired U provides an advantage for cryo-EM study of the Nsp15 and dsRNA complex, which could reveal new structural insights into the Nsp15 binding of the substrate. However, the authors need to clarify whether this new structure unveiled new interactions absent in the previous structures. It would be beneficial to include discussions and figures to compare the new and existing structures and highlight their new findings. These new structural features should then be followed by mutagenesis and assay to confirm their functional importance.

We thank the reviewer for the suggestion to better highlight our new Nsp15 structure and compare it with previously published structures. We have edited the text of our cryo-EM discussion to clarify that the value of our structure is in our experiment design, which would allow us to observe RNA base-specific interactions if they were to exist; our structure thus emphasizes the primarily non-specific nature of Nsp15's interactions with dsRNA, which aligns with and strengthens previously published structures. In addition, we also added a new supplemental figure (S4) in which we compare our structure with the two previously published Nsp15 dsRNA structures. The functional significance of the residues involved in supporting RNA cleavage has been confirmed by mutagenesis in previous studies (Frazier et al NAR 2021; Frazier et al NAR 2022).

Revised text:

Our sequence-defined atomic model shows that most interactions between Nsp15 and the dsRNA are mediated by the phosphate backbone and are not base-specific. **Our model does suggest**

two potential ways in which dsRNA sequence could influence susceptibility to cleavage by Nsp15.

First, we observe discontinuous base stacking and a disruption of H-bonding in the base pair 3' of the flipped U, which presumably help relieve the strain created by the 1 nt bulge unpaired U and imposed by base-flipping; the identity of these bases would influence the ease with which these kinds of distortions to the dsRNA occur.²⁴ Second, our model shows Nsp15 making close contacts with a few of the bases neighboring the flipped U (Figure 3A,C; Supplemental Figure S6, Supplemental Table S3), though these form primarily stacking and hydrophobic interactions. While the identity of these bases may influence the ease with which Nsp15 can position a particular dsRNA around its active site, these interactions are modular rather than strictly sequence-specific. Thus, we propose that a particular dsRNA's susceptibility to cleavage is more likely to be related to qualities of the dsRNA itself (availability of the U, ability of the helix to deform, etc.) rather than any particular base-specific interactions between the dsRNA and Nsp15.

2. A significant emphasis of the present study is the base flipping within dsRNA substrates, where the authors used the ¹⁹F chemical shift of uridines as an indicator for base flipping and stacking. However, the reviewer notes that this approach is too preliminary. Given chemical shifts are highly sensitive to local environments, variations in neighboring nucleotides can substantially affect chemical shifts. For example, the 2'-F chemical shift for Us in the A-U base pair varies by nearly one ppm among sequences studied, as shown in Figure 4. Furthermore, in an RNA sequence different from the ones in the present work, its 2'-F chemical shift of U is downfield shifted when base-paired relative to its unpaired state, opposite to the observations reported here. Hence, to reliably use chemical shifts as an indicator for flipped and stacked Us, the authors need to establish reference chemical shifts for both conformations for each sequence. One suggestion could be measuring chemical shift as a function of temperature, where the unpaired U has a higher likelihood of flipping out at a higher temperature. The authors could also incorporate the Stem Loop 4 sequence, whose U95 is known to flip, to further validate their approach.

In response to the reviewer's suggestion, we clarified our description of our experimental design for Figure 4 – for each sequence, we intended to first compare the chemical shift of unpaired U and paired U-A for a given sequence as a proxy for the comparison between “more likely to be flipped” versus “more likely to be stacked” in the same sequence context. We assigned downfield shifts as being related to flipped character once we confirmed that all unpaired Us across sequences were shifted downfield compared to all U-A sequences.

From our 1D ¹⁹F NMR spectra of dsRNA in the absence of Nsp15 (Figure 4D, Supplemental Table S4), we observe that for each sequence, the peak for the unpaired U is consistently shifted downfield (to the left) relative to the corresponding U-A signal. Though the identity of neighboring nucleotides affects the exact peak location, the range of shifts for unpaired Us does not overlap with those for paired U-As in this system. Previous literature suggests that base flipping events for paired

bases are relatively rare and transitory,^{22,31,48,51} while depending on the sequence, unpaired bases can exist in metastable flipped⁵² or stacked conformations,²⁵ or interconvert readily between flipped and stacked states.⁵³ We therefore hypothesize that the downfield shift of our unpaired Us relative to their corresponding paired U-A signals is at least partially due to exchange with a flipped conformation. In other words, unpaired Us have greater “flipped character” (i.e., spend more time in a flipped or partially flipped conformation) than paired U-As.

Sequences 2 and 5 in particular (“CGUGC” and “AAUAA,” respectively) show an interesting contrast in ¹⁹F NMR: sequence 2 has both the most extremely downfield unpaired U *and* upfield U-A of all sequences we studied, while sequence 5 has unpaired U and paired U-A with the most similar and intermediate shifts. This contrast highlights a trend that is true across all of our sequences –the effect of neighbor context on unpaired Us opposes that on paired U-As. We propose that this contrast may be related to the conformational rigidity of each dsRNA oligo and the resulting ability or inability to accommodate a U in stacking interactions. A more rigid sequence might cause an unpaired U to be locked in a flipped conformation and a paired U-A to be locked in a stacked conformation (e.g., resulting in the strong difference in shifts seen in sequence 2). Likewise, a less rigid sequence might better accommodate the helix distortions required for an unpaired U to occasionally be stacked *and* also allow more frequent base flipping from a paired U-A (e.g., resulting in the intermediate shifts seen in sequence 5). We suspect sequence 5 has lower rigidity around the U than sequence 2 due to its lack of G/C neighbors (fewer H-bonds between each neighboring base pair) and/or the “UUUU” in its complementary strand that would better tolerate discontinuous base stacking.²⁴

We also performed a new set of experiments, as suggested by the reviewer, where we labeled the 2' position of five different Us in SL4 – an unpaired U which is primarily flipped (U95), an unpaired U which is often stacked with its neighbors (U104), a U-A pair (U115), a U•C mismatch (U112), and a U•G wobble pair (U87) – and performed ¹⁹F NMR spectroscopy on each of the five SL4 derivatives. This data has been incorporated into Figure 5. The expanded discussion is included below:

First, we characterized different Us in SL4 by 1D ¹⁹F NMR, to better understand the effect of structure on ¹⁹F NMR shift. To accomplish this, we generated five derivatives of SL4, each with one U labeled with 2'-F (Figure 5A). Based on the ensemble structures of SL4 solved by Vögele et al (Figure 5B),⁵² we labeled an unpaired U which is primarily flipped (U95), an unpaired U which is often stacked with its neighbors (U104), a U-A pair (U115), a U•C mismatch (U112), and a U•G wobble pair (U87) for our ¹⁹F analysis.

The 1D ^{19}F NMR spectra of selected Us in SL4 (Figure 5C) follow the same trends we observed previously in our dsRNA oligos (i.e., in Figure 4). The U-A pair and U•C mismatch show narrow peaks that are shifted upfield (right) relative to the unpaired Us. The unpaired U that is primarily flipped (U95) shows a peak that is shifted strongly downfield (left), while the other unpaired U that is primarily stacked (U104) shows a shift that is remarkably similar to that of the U-A pair. We also note that this unpaired-but-stacked U, which adopts a variety of conformations in the ensemble structures by Vögele et al,⁵² shows the broadest peak of the five Us we labeled. Lastly, the U•G wobble pair is shifted farther downfield than the U-A pair. This validates our hypotheses about the relationships between peak shift and flipped character, peak width and conformational heterogeneity, and bond torsion and peak shift.

And:

Notably, ^{19}F NMR shift is not a perfect predictor of susceptibility to cleavage by Nsp15, with U112 being cleaved more and U87 less than would be expected based purely on shift. We emphasize the importance of controlling for sequence and secondary structure context when making comparisons between ^{19}F shifts, as ^{19}F shifts are sensitive to multiple confounding stimuli that can challenge interpretation. U112, for example, is adjacent to a U•G wobble pair that may affect bond torsions in dimensions not experienced by any other U we probed.

*We note that there are two papers to which the reviewer may be referring (where ^{19}F NMR of 2'-F-labeled U appears to contradict the trends we observe with our substrates). One paper (Dow et al JACS 2019) observes upfield shift of U on flipping **in the presence of enzyme in the context of DNA repair**; the upfield shift here is likely related to the chemical environment within the enzyme active site, which is distinct from our experiment where a flipped base is more exposed to solvent than a stacked base. This paper does not report flipped versus stacked U ^{19}F NMR shifts in the absence of enzyme, only stacked (no enzyme) versus flipped (into enzyme active site). The second paper (Kreutz et al JACS 2005) use ^{19}F NMR to probe the structure of a bistable RNA that can exchange between two hairpin-with-tail conformations. This paper reports 2'-F-U shifting upfield when it is "single stranded" compared to when it is involved in the "double stranded" helix of a hairpin; however a base **being "single stranded" does not exclude the base from stacking interactions**, as we observe in U104 of SL4 in our data, **and does not necessarily increase solvent-accessibility**. We have added the following clarification to our explanation of our ^{19}F NMR experiments.*

This technique has previously been used to probe enzymatic capture of flipped bases in DNA,⁴⁸ as well as interconversion between two conformations in a bistable RNA hairpin,⁴⁹ but has not yet been used to probe spontaneous base flipping in dsRNA.

3. Based on the data from the Stem Loop 4, the authors suggested that cleavage by Nsp15 at one site within dsRNA can increase the subsequent reactivity of nearby bases. Since U95 is the preferred cleavage site for Nsp15, the increased cleavage at U120 following U95 cleavage could potentially be attributed to U120 becoming the only cleavage site available for Nsp15 rather than an actual increase in its intrinsic cleavage rate. Could the authors provide further elaboration on their proposed mechanism?

We have added an explanation of this phenomenon to this section:

This chain reaction of enhanced reactivity is in line with our observations that Us with greater solvent accessibility are more reactive to Nsp15 – each cleavage event introduces nicks that increase the flexibility of the dsRNA,²⁴ making a new set of bases more accessible to Nsp15. We suspect both of these secondary cleavage sites (C100 and U120) were singled out by Nsp15 after SL4 had been nicked elsewhere because of greater than average flexibility at these bases: C100 is complementary to one of the primary cut sites (U112), and U120 is adjacent to a C•C mismatch and only 3 base pairs away from the most favored primary cut site (U95).

4. It would benefit the readers if the authors could quantify the cleavage assays presented in Figs 2 and 5, similar to the quantification provided in Figs 1 and 4. This will provide consistency in data analysis and presentation.

We refrained from quantifying the assays in Figs 2 and 5 to avoid conflating these assays (on non-fluorinated RNA, with multiple potential cleavage targets) with the assays with uncleavable 2'-F-U (cleavage focused on one target). Quantifying Figs 1 and 4 allows us to make very specific comparisons between Us in slightly different contexts. By contrast, the multiple cleavage targets readily available in substrates for Fig 2 means disappearance of uncleaved RNA cannot be directly attributed to cleavage at the target U. Also, intensity of product bands is not a full picture of cleavage at the target U(s), especially for Figs 2 and 5, due to subsequent cleavage reactions that occur to all cleavage products.

We have added the following to our Methods to clarify this point:

For substrates containing 2'-F-U where cleavage was focused on a single target U (Figures 1 and 4, and Supplemental Figure S8), RNA cleavage was quantified for each reaction by measuring the decreasing intensity of the intact RNA band in both red and blue channels, performing a baseline correction by subtracting the intensity of an empty well, then normalizing to the baseline-corrected intensity of the t = 2 min timepoint (setting this intensity equal to "0% cleaved"). Measurements of % cleavage were averaged across red and blue channels for at least three biological replicates. Quantifying decrease in uncleaved RNA intensity was determined to be the least noisy reporter for susceptibility to cleavage by Nsp15 for these assays (as opposed to quantifying increase in product signal), since cleavage products can undergo subsequent digestion at off-target sites, including

sometimes C instead of U. For this reason, reactions with native oligos (containing no 2'-F-U, i.e.,

Figures 2 and 5, and Supplemental Figure S9) were not quantified.

5. The authors discussed the linewidth of the ¹⁹F NMR data in the manuscript. However, it was unclear where these data were specifically presented, presumably the values in parentheses in Figure 4. To improve the clarity, the authors need to reference these data properly and include detailed information in the figure legend.

Supplemental Table S4 has been added to the Supplemental Information listing all peak shifts and linewidths. Additionally, the following has been added to the captions of Figures 4 and 5.

Each peak is labeled with its shift (top value) and linewidth (bottom value, in parentheses).

Reviewer #2 (Remarks to the Author):

The manuscript addresses the question of whether the endonuclease Nsp15 targets U residues that are already flipped or whether it induces flipping. The authors combine in vitro assays, cryoEM, and NMR analysis of U conformation to provide the answer and reveal important principles of specificity for an important covid enzyme.

There is a fair amount of qualitative data in the literature identifying some of the basic characteristics of preferred substrates for Nsp15. However, quantitative data regarding alternative substrates that have been carefully designed to test specific features of RNA structure are very much lacking. Moreover, several studies have now shown that RNA structure is important for Nsp15 specificity. Both the protein and RNA structure are dynamic making this aspect another key issue to address for the function and specificity of this enzyme.

Therefore, this manuscript represents an important step forward since it builds on both of these areas where mechanistic detail is needed. The data establish new insights into structure specificity of Nsp15 and are sufficiently rigorous and quantitative to provide benchmarks for more detailed study of this class of enzymes.

In my view there are two important issues (1 and 2 below) that need to be resolved and three additional minor comments (3-5) to consider.

We thank the reviewer for their supportive comments and suggestions.

1. The authors need to include the substrate and enzyme concentrations in the text and figure legends. They need to introduce and discuss the rationale for selecting these the concentrations of E and S and rationalize their results in terms of whether the way they set up their reactions will reflect differences between alternative sites due to catalysis, binding, cooperativity, or some other aspect of the reaction.

The following was added to the caption of Figure 1:

B) Representative denaturing PAGE gels of timecourse nuclease assays for each of the five substrates, performed at 50 nM Nsp15 and 500 nM dsRNA.

Also, the following was added to the Methods:

To perform one reaction, Nsp15 (50 nM) was incubated with dsRNA (500 nM) at room temperature in assay buffer (20 mM HEPES pH 7.5, 150 mM NaCl, 5 mM MnCl₂, 5 mM DTT, and 1 U/ μL RNasin) for 60 min. **These concentrations of enzyme and substrate were previously optimized^{8,11} for observing differences in cleavage between dsRNA substrates over the course of 1 hr.**

While kinetics analysis of Nsp15 with dsRNA would be a valuable addition to the field, it is outside the scope of this manuscript. In fact, to effectively answer our questions about the relationship between dsRNA sequence, base flipping, and susceptibility to cleavage by Nsp15, we had to make design choices for our oligos that are incompatible with true kinetics analysis, for several reasons:

- First, the length of our dsRNA oligos (~35 nts) was selected to capture the global and long-range interactions that influence base flipping and distinguish it from chain end fraying. We also opted to include multiple Us in the sequence because a long dsRNA sequence with extreme G/C content would be difficult to properly anneal and have unrealistic properties compared to Nsp15's typical substrates. Having multiple, non-equivalent targets in each dsRNA oligo severely complicates true kinetic modeling of our system.*
- Second, our 2'-F-U dsRNA substrates contain multiple Us that can be sampled by Nsp15's active site without being cleaved. Thus, our assay can only observe a subset of the endoU-dsRNA binding events taking place, which challenges a comprehensive kinetic analysis of the whole system, but strategically allows us to compare the susceptibility of a particular target U in different local contexts **against the same background sequence**. (Our % cleavage data represents a ratio of how often the target U is sampled by the active site i.e. cleaved compared to how often all other Us are sampled but not cleaved. It would be inappropriate to broadly compare cleavage of 2'-F-U dsRNA oligos with dramatically different sequences.) See next comment and response for how we addressed this in the text.*
- Third, Nsp15 is active as a hexamer. Previous kinetic (Huang et al JBC 2023) and structural analyses (Jernigan et al Structure 2023; Godoy et al NAR 2023) suggest that there is some level of cooperativity between the six different active further complicating the proper kinetic analysis of dsRNA cleavage.*

2. There is an apparent inconsistency between the data in Figure 4 and the model for Nsp15 recognition in Figure 6. Several substrates (e.g. #1, 4 and 50) do not react to completion (only 10-50%). This effect is independent of the rate based on the data shown in Figure 4. However, the mechanism shown in Figure 6 predicts that the entire population of RNA should be cleaved because the different structural forms of the substrate are in equilibrium. In this model all RNAs will necessarily sample the U flipped conformation and eventually become a substrate. Are the authors claiming that flipping is so slow that there is no exchange between RNA conformations over the time course of the reaction? Does this conclusion follow from the NMR data? Some additional explanation is needed to resolve this apparent conflict between the data and the model.

This effect is a quirk of our substrates that contain uncleavable 2'-F-U's. We have added this to our explanation of the data in Figure 1:

We note that in these assays, dsRNA cleavage plateaus over the course of the hour for all substrates, even those where total cleavage remains low. We attribute this to the presence of 2'-F-U's in the target strand, which continue to be sampled by Nsp15's active site (potentially at a greater frequency than the target U) but are not able to undergo cleavage.

And this to our Methods:

Measurements of % cleavage were averaged across red and blue channels for at least three biological replicates. Because 2'-F-U's can be sampled by Nsp15's active site but not cleaved, the % cleavage represents a ratio of how often the target U was sampled (and cleaved) relative to how often other Us in the sequence were sampled.

3. In Figure 1C, does the U-U mismatch substrate have a 2'-F-U modification on the non-labeled strand? This seems to follow from the overall description of the substrate design, but it would be useful for the authors to clarify this since it impacts the interpretation to some degree.

Originally, our U•U mismatches did not use a 2'-F-U on the complimentary (non-labeled) strand. We have performed a new set of experiments using a U•(2'-F-U), and swapped this 2'-F data into Figure 1. We have moved the original U•U data to the Supplemental Information as Supplemental Figure S1, updated statistical analyses to include the new U•(2'-F-U) data, and edited the following text.

To directly compare cleavage between substrates, all Us other than U19 in the target strand were replaced by 2'-fluoro-uridines (2'-F-U's), which can be sampled by Nsp15's active site but cannot undergo the chemical reaction necessary for cleavage. For the U•U' mismatch, we also substituted the complementary U for an uncleavable 2'-F-U (U').

...

The U-A match, U•U' mismatch, and U•C mismatch were intermediate, achieving $28 \pm 7\%$, $25 \pm 5\%$, and $21 \pm 7\%$ cleavage respectively. A one-way ANOVA followed by a Tukey HSD post-hoc test indicates that cleavage of the unpaired U is significantly greater than all other substrates ($p = 0.0002$, 0.0002 , 0.0001 , 0.0000 for unpaired U versus U-A, U•U', U•C, U•G respectively with a 95% confidence interval) and cleavage of U•G is significantly less than cleavage of all other substrates except for U•C ($p = 0.0000$, 0.0065 , 0.0311 , 0.1220 for U•G versus unpaired U, U-A, U•U', U•C respectively).

...

We also tested susceptibility to cleavage for an alternate U•U mismatch, where both the target U19 and its complementary U are cleavable (Supplementary Figure S1). We found that the target U19 for this substrate was slightly more susceptible to cleavage than the U•U' mismatch discussed above ($37 \pm 8\%$ vs. $25 \pm 5\%$), but not enough to be statistically significant. Nicking at the complementary U in the U•U mismatch may contribute to the target U19's increased cleavability,^{12,24} though our assay does not have the statistical power to determine this with certainty.

4. The conclusion that bulged U is a key determinant for Nsp15 cleavage is important. However, an excess of dsRNA binding sites will necessarily reduce the rate of cleavage at a small number of bulged sites. In the discussion the authors might mention that the concentrations of alternative substrates in addition to intrinsic cleavage rate also determines relative cleavage site preference in vivo.

We added the following to our conclusion:

Ultimately, Nsp15's cleavage activity during infection will be influenced by its association with other proteins in the double membrane replication vesicle, which likely mediates its access to viral RNA and may additionally affect RNA structure. The relative abundance of bulged versus paired Us in target RNA would also influence Nsp15's activity in vivo, with the potential to funnel cleavage towards less preferred substrates under the circumstance that more preferred Us are dramatically outnumbered. (The magnitude of this effect would depend on the frequencies and lifetimes of base flipping events toward the minor groove for all base pairs present.)

5. There are some minor typos in lines 461 and 462

These typos have been corrected.

Reviewer #3 (Remarks to the Author):

Reviewer #4 (Remarks to the Author):

• What are the noteworthy results?

This really interesting paper by Wright and coauthors investigates the really important question on the mechanism of dsRNA cleavage of the SARS-CoV2 protein NSP15 (alternate name NendoU), which is a key enzyme responsible to “hiding” the presence of the virus from the host immune system. It is present in all coronaviruses. The authors use a very impressive method spectrum, including functional cleavage assays, Cryo-EM, NMR, and mass spectroscopy to

investigate the hypothesis that base-flipping of the U substrate is essential for the cleavage mechanism. They designed different dsRNA substrates, where they systematically varied the base pairing of the “substrate U” as well as the surrounding bases and also investigated the cleavage preference of one of the potential native substrates, the Stem Loop 4 from the SARS-CoV-2 genome. They discovered that the Nsp15’s cleavage efficiency in dsRNA is driven by the accessibility of the U. They were able to show that Nsp15 cleaves unpaired Us most efficiently but can also cleave Us in a range of complement, neighbor, and secondary structure contexts. Here some initially unexpected results were obtained, for example, that cleavage still occurs, when the U is fully base paired with A, but cleavage is nearly completely abolished in the case of a U-G wobble pair. In the discussion, they presented a very convincing interpretation of this unexpected result: The catalytic site of Nsp15 will only be able to access and cleave base flipped U when it is flipped into the minor groove of the dsRNA, which is the preferential orientation in unpaired U and also fully base-paired U-A but in the case for the U-G wobble pair U is flipped in the major groove of the ds-RNA and can therefore not be cleaved by Nsp15. They also varied the neighboring bases of the U-A in the dsRNA substrates and discovered that there is a high cleavage efficiency when the paired U-A is neighbored by Us, which could be explained that the presence of neighboring Us leads to a lower energetic barrier to base unstacking compared to a paired U-A neighbored by As or Gs. The authors then investigated the conformation of the different substrates (without Nsp15) by ¹⁹F NMR where they found that % cleavage of a U by Nsp15 is correlated with the peak shift of 2'-F-U in ¹⁹F NMR, which is directly related to solvent accessibility of the 2'-F. The NMR data were correlated very well with the results on the cleavage as the peak shift is most pronounced in dsRNA with unpaired U. Finally, they were able to show that Nsp15 also strongly prefers to cleave the unpaired U95 in SL4 over all other Us, where previous studies have shown that U85 is fully extruded from the helix and solvent-accessible. The cleavage essays and NMR experiments are further supported by a cryo-EM structure of Nsp15 at medium resolution, where they solved the structure of Nsp15 in complex with the same dsRNA with an unpaired U that was used for the functional studies shown in Figure 2. They identified the dsRNA in the binding groove of subunit P1 of the hexameric Nsp15 and proposed a network of amino acids that interact with the dsRNA and specifically the base flipped U in the minor groove of the dsRNA, which is slightly widened in the structure of the Nsp15-dsRNA complex.

In conclusion, this exciting paper shows that Nsp15 does not actively base-flip the U but binds to and efficiently cleaves dsRNA substrates where U is in a base-flipped conformation. The widening of the minor groove could be induced by Nsp15 and thereby could contribute to the stabilization of the base flipped conformation and thereby could explain the efficient cleavage of fully base paired U-A where U base flipping is only a transient event.

• Will the work be of significance to the field and related fields? How does it compare to the established literature? If the work is not original, please provide relevant references.

The work is of very high significance to the field as it presents a very important finding of the mechanism of the cleavage of dsRNA by Nsp15. The work presented here is original and the conclusions are supported by a very carefully selected spectrum of methods ranging from the systematic design of dsRNA substrates to functional cleavage assays which are combined with structural studies by Cryo-EM as well as NMR studies on the dsRNA substrates. The authors relate their work to the existing literature leading to a very convincing picture of the mechanisms of dsRNA cleavage by Nsp15. It was especially impressive that their studies on the stem-loop 4 showed that Nsp15 cleaves this substrate nearly exclusively at U95 site which has been shown to be fully base flipped in previous studies.

The results presented here and the conclusions are of extremely high relevance to understanding the cleavage mechanism and may also have a huge impact on the development

of potential future drugs that could inhibit the Nsp15 function and thereby prevent SARS-CoV-2 (and future coronaviruses that will become threats for human health) to hide the virus from the immune system.

- Does the work support the conclusions and claims, or is additional evidence needed?

The work presented here fully supports the exciting conclusions about the mechanism of NendoU “preying on based flipped U presented in this paper.

- Are there any flaws in the data analysis, interpretation, and conclusions? Do these prohibit publication or require revision?

The analysis of the data is done very carefully and the reviewer has not identified any flaws in the data analysis and interpretation. I have some comments and suggestions cc the representation of the Cryo-EM data in the Figures listed below, which should be revised to **show more details of the interaction of the dsRNA and especially the base flipped U with the amino acids in the catalytic site, which should be shown not just as a stick model but showing also the experimental density map for both the protein and the dsRNA.** This revision should be included in the revised version of the paper.

- Is the methodology sound? Does the work meet the expected standards in your field?

The methodology is solid. All experiments are carefully designed and executed. The reviewer also really appreciated that the authors tell a very convincing story in their paper where they walk the reader through the reasoning for each of the consecutive experiments conducted so that the reader can follow nicely the experimental plan and see how the results of one experiment informed the experimental plan for the next step.

- Is there enough detail provided in the methods for the work to be reproduced?

The methods section of the paper is very detailed and describes all experiments in sufficient detail including isolation of the Nsp15 WT and its non-active mutant (used for the Cryo-EM studies), which can bind the substrate but cannot cleave it.

We thank the reviewer for their supportive comments and helpful suggestions.

Below you will find some more minor comments and suggestions that should be included in the revision of the manuscript:

Cryo-EM structure analysis and interpretation:

The authors show multiple similar overview figures of the structure of Nsp15 bound to the dsRNA substrate but only one figure (Figure 3 D) also shows the experimental density for the small 6 nucleotide section of the dsRNA that includes the flipped U14 base. While this density looks very convincing none of the figures show experimental densities for the protein. As the authors include a detailed discussion on the amino acids that interact with the dsRNA this should be supported by **showing the density maps for key residues that foster the protein dsRNA interactions. These maps with the side chain densities should be included in the supplementary figures S2 and S3.** It would be also helpful to include a **table with the shortest edge-to-edge distances of key residues with the dsRNA** in the supplementary information.

Supplemental Figure S3 has been updated to include the cryo-EM density maps for residues of Nsp15 that foster interactions with dsRNA, and the figure caption has been updated appropriately. We have also added a new Supplemental Figure (S6), which highlights base-specific interactions (i.e., interactions between Nsp15 and bases, excluding interactions with the RNA's sugar phosphate backbone), and Supplemental Table S3, which lists the distances between key Nsp15 and dsRNA residues that have relatively strong interactions.

Further suggestions on Figures:

Supplementary Figure S4

Panels C and D: the light and dark blue colors are nearly indistinguishable. Please choose a different color scheme here.

We have introduced unique symbols for each of our substrates throughout the paper, and added labels to figures to indicate this. This should enhance legibility of the figures even when readers are viewing in greyscale.

Panel E: The authors have indicated that all experiments were performed in triplicates. Please include error bars in this figure derived from the triplicate experiments.

X error bars have been added to this Supplemental Figure (now S8) as well as the corresponding panel E in the main text Figure 4. The captions for these figures have been updated appropriately.

E) Scatter plot of shift (peak position) from 1D ¹⁹F NMR spectra vs. % cleaved at 60 min from our enzymatic assays. For each point, x position represents the average of three independent reactions for a single substrate (error bars represent standard deviation), y position represents one spectrum.

Figures 1 and 2 (gels) and the interpretation of the cleavage results

The authors have based their analysis of the % cleavage on the scanning of the intensity of the non-cleaved substrate, instead of the appearance of the cleavage products and the reviewers wonder why as the appearance of the cleavage products is much more prominently visible in all gels compared to the diminishing of the non-cleaved dsRNA. Looking at Figure 1 specifically there is a discrepancy between the time course shown in Figure 1C (based on the disappearance of the dsRNA band) and the time course of the appearance of the cleaved products for the A-H fully base-paired sample and U-U mismatch. The interpretation of the results based on the dsRNA band shows an order of cleavage efficiency of U>U-U mismatch >U-A match>U-C mismatch >U-G wobble. However, when following the appearance of the cleaved bands the order is different, where the cleavage efficiency of the U-A match is second to the unpaired U. For the unpaired U cleavage products are clearly visible at 5 min while cleavage products for the U-A match already prominently appear already just 5 min later at 10 min while the cleavage bands for U-U and U-C do not appear before 15 and 30 minutes. This order where the U-A match is faster and more efficiently cleaved compared to the mismatches U-U and U-C is also visible from the intensity of the cleavage bands which are more prominent for U-A than for U-U and U-C. It would be important that the author explain why they do not use the cleavage products for the interpretation of the cleavage efficiency and why the order of cleavage efficiency is different when looking at the cleavage products.

We have found quantifying an increase in cleavage product to be a noisier readout for this type of assay, since cleavage products can continue to undergo secondary digestion by Nsp15, particularly since each cleavage event makes new cleavage sites more accessible. Our assay is interested in directly comparing specific cleavage events in oligos with a controlled structure, so quantifying the decrease in uncleaved RNA where off-target events are less likely to occur is more appropriate here. (Nsp15 can sometimes cleave at C instead of U, so even our 2'-F-U substrates in Figures 1 and 4 are not immune to this secondary digestion. Substituting all Cs in our oligo for uncleavable 2'-F-Cs in addition to having multiple 2'-F-Us would have substantially affected the thermal stability of our oligos, compromising the assay.)

We have added the following to address this in the manuscript:

Quantifying decrease in uncleaved RNA intensity was determined to be the least noisy reporter for susceptibility to cleavage by Nsp15 for these assays (as opposed to quantifying increase in product signal), since cleavage products can undergo subsequent digestion at off-target sites, including sometimes C instead of U.

We note that there is no discrepancy between the order of cleavage efficiency whether looking at decrease in uncleaved RNA or increase in cleavage product. The reviewer mentions that they do not see bands for product appear for U•U and U•C before 15 min, but there are product bands present even at 2 min for all substrates except the resistant U•G wobble, though they are somewhat faint. We also stress that we aimed to select representative gel images that clearly reflect the trends we discuss, but that there is some variation between each biological replicate, which is captured by the error bars in Figure 1C.

REVIEWERS' COMMENTS

Reviewer #1 (Remarks to the Author):

The revised manuscript by Wright et al. has addressed most of my concerns from the original manuscript. However, there are a few minor errors in the revised version, specifically where the 5' and 3' labels on the RNA secondary structures are reversed, and the chemical shifts should be in units of ppm rather than Hz. With these corrections, the reviewer recommends this work for publication in Nature Communications.

Reviewer #2 (Remarks to the Author):

The authors have carefully considered the major issues I raised in my opinion adequately responded to the others as well.

They provide new supplementary figure comparing current structure to previous nsp15 ES complexes which is an important addition.

Authors adequately address key issues raised regarding the assignment and interpretation of NMR signals to different conformational states, i.e relationship between flipped character and and NMR peak shift.

The authors thoughtfully address my comments regarding putting the results in context of what is known about Nsp15 kinetics and clarified the issue with reactions not proceeding to completion in Figs 4 and 6.

Reviewer #3 (Remarks to the Author):

Reviewer #4 (Remarks to the Author):

Reviewer comments on the revised manuscript of Wright et al on Nsp15 can't take its eyes off unpaired U: spontaneous base flipping helps drive Nsp15's preferences in dsRNA substrates

- The reviewer is very satisfied with the revised version of Wright et al. 's manuscript. The authors have very carefully revised the manuscript based on the comments of all 3+1 reviewers and explained the changes in their answers to the reviewers. The reviewer also highly appreciates that they have highlighted the changes in the manuscript directly in the point-to-point answer to the reviewer's comments.
- The manuscript now includes very detailed answers including new data in response to reviewers 1 and 2. They have also explained why they have not attempted detailed kinetic analysis as suggested by reviewer 2. The major reason is that the kinetics of the cleavage by hexameric NSP15 is complex due to allosteric regulation, which has been shown previously in the literature.
- The authors also have fully addressed the comments of this reviewer (reviewer 4).
- The figures have been revised and the density for the protein is now shown in the structure figures. Supplemental Figure S3 has been updated to include the cryo-EM density maps for residues of Nsp15 that foster interactions with dsRNA, and the figure caption has been updated appropriately. They have also added a new Supplemental Figure (S6) highlighting base-specific interactions.
- Error bars have been added to the Supplemental Figure (now S8) as well as the corresponding panel E in the main text Figure 4 as requested by the reviewer.
- The authors have also convincingly explained why they use the reduction of the original RNA and not the appearance of the cleavage products as the basis for the determination of the cleavage efficiency as secondary cleavage products including cleavage after C could occur which would complicate the interpretation of the results. They now mention this explanation in the text.
- As outlined in the original review, the reviewer finds this paper extremely important and is very enthusiastic about the exciting results presented in this paper. The reviewer is fully satisfied with the revisions and strongly recommends the publication of the paper in its revised form in Nature Communications

Rebuttal NCOMMS-24-21724

Reviewer comments (black text) and our response (*blue italics*)

Reviewer #1 (Remarks to the Author):

The revised manuscript by Wright et al. has addressed most of my concerns from the original manuscript. However, there are a few minor errors in the revised version, specifically where the 5' and 3' labels on the RNA secondary structures are reversed, and the chemical shifts should be in units of ppm rather than Hz. With these corrections, the reviewer recommends this work for publication in Nature Communications.

We thank the reviewer for their supportive feedback and for catching these minor errors. We have corrected the 5' and 3' labels on the RNA secondary structures in Figure 5 and the chemical shifts have been updated to ppm.

Reviewer #2 (Remarks to the Author):

The authors have carefully considered the major issues I raised in my opinion adequately responded to the others as well.

They provide new supplementary figure comparing current structure to previous nsp15 ES complexes which is an important addition.

Authors adequately address key issues raised regarding the assignment and interpretation of NMR signals to different conformational states, i.e relationship between flipped character and NMR peak shift.

The authors thoughtfully address my comments regarding putting the results in context of what is known about Nsp15 kinetics and clarified the issue with reactions not proceeding to completion in Figs 4 and 6.

We thank the reviewer for their supportive comments.

Reviewer #3 (Remarks to the Author):

Reviewer #4 (Remarks to the Author):

Reviewer comments on the revised manuscript of Wright et al on Nsp15 can't take its eyes off unpaired U: spontaneous base flipping helps drive Nsp15's preferences in dsRNA substrates

- The reviewer is very satisfied with the revised version of Wright et al. 's manuscript. The authors have very carefully revised the manuscript based on the comments of all 3+1 reviewers and explained the changes in their answers to the reviewers. The reviewer also highly appreciates that they have highlighted the changes in the manuscript directly in the point-to-point answer to the reviewer's comments.
- The manuscript now includes very detailed answers including new data in response to reviewers 1 and 2. They have also explained why they have not attempted detailed kinetic

analysis as suggested by reviewer 2. The major reason is that the kinetics of the cleavage by hexameric NSP15 is complex due to allosteric regulation, which has been shown previously in the literature.

- The authors also have fully addressed the comments of this reviewer (reviewer 4).
- The figures have been revised and the density for the protein is now shown in the structure figures. Supplemental Figure S3 has been updated to include the cryo-EM density maps for residues of Nsp15 that foster interactions with dsRNA, and the figure caption has been updated appropriately. They have also added a new Supplemental Figure (S6) highlighting base-specific interactions.
- Error bars have been added to the Supplemental Figure (now S8) as well as the corresponding panel E in the main text Figure 4 as requested by the reviewer.
- The authors have also convincingly explained why they use the reduction of the original RNA and not the appearance of the cleavage products as the basis for the determination of the cleavage efficiency as secondary cleavage products including cleavage after C could occur which would complicate the interpretation of the results. They now mention this explanation in the text.
- As outlined in the original review, the reviewer finds this paper extremely important and is very enthusiastic about the exciting results presented in this paper. The reviewer is fully satisfied with the revisions and strongly recommends the publication of the paper in its revised form in Nature Communications

We thank the reviewer for the supportive comments on our revised manuscript.